# Analysis of hygroscopic cloud seeding materials using the Korea Cloud Physics Experimental Chamber (K-CPEC): A case study for powder-type sodium chloride and calcium chloride

Bu-Yo Kim[1], Miloslav Belorid[1], Joo Wan Cha[1], Youngmi Kim[1], Seungbum Kim[1]

[1]Research Applications Department, National Institute of Meteorological Sciences, Seogwipo, Jeju 63568, Republic of Korea

*Correspondence to*: Bu-Yo Kim (kimbuyo@korea.kr)

**Abstract:** In this study, we analyzed the particle characteristics and cloud droplet growth properties of NaCl and $CaCl_2$, which are powder-type hygroscopic materials applied in cloud seeding experiments, using the Korea Cloud Physics Experimental Chamber (K-CPEC) facility at the Korea Meteorological Administration/National Institute of Meteorological Sciences (KMA/NIMS) in South Korea. The aerosol chamber (volume 28.3 $m^3$) enabled the observation of particle characteristics in an extremely dry environment (relative humidity (RH) < 1 %) that was clean enough to ignore the influence of background aerosols. The cloud chamber featured a double-structure design, with an outer (130 $m^3$) and inner (22.4 $m^3$) chamber. The inner chamber allowed the precise control of air pressure (1013.25–30 hPa) and wall temperature (–70–60 °C), facilitating cloud droplet growth through quasi-adiabatic expansion. In this study, a cloud chamber experiment was conducted to simulate both wet adiabatic and stable environmental lapse rate conditions. The experiments were initiated at low RH (< 60 %), and the variations in the cloud droplet concentration and diameter were observed as RH increased, leading to supersaturation (RH > 100 %) and subsequent cloud droplet formation. NaCl and $CaCl_2$ powders showed distinct particle growth behaviors owing to the differences in their deliquescence and hygroscopicity. The rate of cloud droplet formation in the NaCl powder experiments was slower than that for $CaCl_2$; however, the mean and maximum droplet diameters were approximately 2–3 μm and 10–20 μm larger, respectively. The particle diameter, including aerosols and droplets, varied from 1 to 90 μm, and large cloud droplets (30–50 μm) that served as the basis for drizzle embryo formation were also observed. Our study provides valuable insights for the development of new seeding materials and advanced cloud seeding experiments.

## 1 Introduction

The rise in the temperature of the Earth's surface and atmosphere is causing climate change and increasing the intensity and frequency of meteorological disasters, such as heavy rains, floods, heat waves, and droughts, which have severe repercussions for life and property (Kim et al., 2020a; IPCC, 2022; Kim and Cha, 2025). In addition, climate change and rising temperature cause higher evapotranspiration in soil and plants, affecting the hydrological cycle and water resources (Roche et al., 2018; He et al., 2022). In dry areas, more than 90 % of the annual precipitation can be released into the atmosphere (Schneider et al., 2021). Increased evapotranspiration and intensified hydrological cycles can change the availability of water resources,

resulting in increased vegetation stress, land cover change, and wildfires (Koppa et al., 2022; Sezen, 2023). Therefore, several countries are implementing measures to secure water resources and prevent natural disasters (Kim et al., 2024). In particular, the interest in eco-friendly and economical weather-modification technologies (e.g., cloud seeding) for precipitation enhancement is increasing, boosting the demand for the research and development of these techniques.

Cloud seeding is used to induce precipitation (in the form of snow, droplets, or ice) in clouds with low-precipitation

probability or efficiency using artificially implemented microphysical processes (Silverman, 2001). This technique involves the use of seeding materials that perform the role of condensation nuclei and ice nuclei, which are sprayed around clouds to enhance the collision-coalescence or deliquescence-heterogeneous freezing processes of clouds, resulting in efficient and effective precipitation (Bruintjes, 1999; Khvorostyanov and Curry, 2004). Several countries, including China, Thailand, and the United Arab Emirates (UAE), and the United States of America (USA), have demonstrated an increase in annual

precipitation through cloud seeding, using meteorological aircraft, drones, unmanned aerial vehicles (UAVs), rockets, and ground-based aerosol generators (Flossmann et al., 2019; Wondie, 2023). Weather-modification technologies are being developed and commercialized, with several economic benefits being reported worldwide (Tessendorf et al., 2019; Kim et al., 2020b; Knowles and Skidmore, 2021). Cloud seeding can shorten the duration of the precipitation process and increase precipitation intensity. However, it may not always result in more rainfall than natural precipitation, depending on

meteorological conditions (Silverman, 2003). Therefore, proper assessment of meteorological conditions and appropriate seeding strategies are essential for effective cloud seeding.

In South Korea, after developing the Korea Meteorological Administration (KMA)/National Institute of Meteorological Sciences (NIMS) Atmospheric Research Aircraft (NARA, aircraft type: Beechcraft King Air 350 HW), cloud seeding experiments have been conducted in since 2018 for inducing artificial rains, preventing forest fire, promoting fog

dissipation, and reducing fine dust (Cha et al., 2019; Kim et al., 2020a; Lim et al., 2022; Ku et al., 2023). Cloud seeding using a meteorological aircraft involves the use of burn-in-place flare type $CaCl_2$ for warm clouds (above –5 °C) and AgI flare for cold clouds (below –5 °C) (Rosenfeld et al., 2005). Since 2022, in collaboration with the Republic of Korea Air Force (ROKAF), cloud seeding experiments have been conducted using an Air Force transport aircraft (CN235) to seed warm clouds with powder-type hygroscopic materials (such as NaCl and $CaCl_2$) (Lim et al., 2023; NIMS, 2023). The flare-type operation

disperses seeding materials at a constant rate once ignited, limiting the ability to adjust the quantity during cloud seeding experiments. The powder-type operation—provided that sufficient cargo space is available within the aircraft—allows for more precise control over both the quantity and rate of seeding. It also accommodates a wider range of seeding agents and is relatively more cost-effective than flare-type agents. In addition, novel cloud-seeding technologies that employ rockets, drones, and UAVs are being developed for conducting cloud seeding using a ground-based aerosol generator in a mountainous area

(Daegwallyeong) at an altitude of 772 m (Jung et al., 2022; Cha et al., 2024; Koo et al., 2024).

The mean annual precipitation in South Korea is 1300 mm, with 70 % of the annual precipitation being concentrated to the rainy season (May–October) and more than 50 % of the precipitation occurring in summer (June–August) (Park et al., 2021). In Korea, droughts tend to occur in summer, with severe droughts lasting until the winter of that year or the following

summer (Ham et al., 2024). Insufficient precipitation during the rainy season may lead to drought in winter (Kim et al., 2020b).

Therefore, continuous cloud-seeding experiments are required to secure water resources. The clouds that are suitable for cloud seeding in South Korea are characterized are low-altitude clouds (stratocumulus (Sc) and cumulus (Cu)), with mean frequencies of 63 % and 15 % in all seasons, respectively, and middle-altitude clouds (altostratus (As) and altocumulus (Ac)), with mean frequencies of 13 % and 6 %, respectively (Kim et al., 2020a). In general, the mean cloud top height of low-altitude clouds is 2.42 km, and the mean cloud top temperature is 1.15 ℃; for middle-altitude clouds, the mean cloud top height is 3.27

km, and the mean cloud top temperature is –1.80 ℃. During the period from spring to fall, including the rainy season, similar cloud frequencies (Sc: 63 %, Cu: 15 %, As: 15 %, and Ac: 7 %), cloud top heights (low-altitude: 2.45 km and middle-altitude: 3.26 km), and relatively high cloud top temperatures (low-altitude: 4.01 ℃ and middle-altitude: 0.61 ℃) were observed. Thus, the cloud temperature and altitude conditions in South Korea are suitable for conducting cloud seeding experiments for warm clouds from spring to fall.

The effectiveness of cloud seeding is closely related to the characteristics of the seeding material (e.g., type, diameter, size distribution, and number concentration) and the meteorological conditions of the cloud (e.g., air temperature, saturation, and updraft velocity) (Hoppel et al., 1994; Li et al., 2023). The effectiveness of cloud seeding experiments can be verified using in situ radar, satellite, ground, and airborne data and numerical model results (Tessendorf et al., 2019). However, the interactions between different meteorological factors in the atmosphere at every moment are complex. Therefore, the

observation and analysis of the growth of water droplets and ice crystals and the assessment of the effectiveness of cloud seeding based on the characteristics of the seeding material involve high uncertainty (Schneider et al., 2021). Several cloud chambers around the world, including the Aerosol Interactions and Dynamics in the Atmosphere (AIDA) facility in Germany (Wagner et al., 2006), Pi (Π) in the USA (Chang et al., 2016), Manchester Ice Cloud Chamber (MICC) in the United Kingdom (UK) (Shao et al., 2022), Big Cloud Chamber (BCC) in Russia (Drofa et al., 2010), Beijing Aerosol and Cloud Interaction

Chamber (BACIC) in China (Li et al., 2023), and Meteorological Research Institute (MRI) in Japan (Tajiri et al., 2013), were constructed in order to conduct extensive research on aerosol-cloud-precipitation-climate interactions (Shaw et al., 2020). Although all chambers have their limitations, they can repeatedly simulate various experiments in environments similar to those of cloud seeding experiments using aircraft, allowing for the exploration of conditions that may be difficult to replicate or observe during flight.

The Korea Cloud Physics Experimental Chamber (K-CPEC) facility in South Korea was first built to develop cloud seeding technology to prevent natural disasters (wildfires and droughts) and secure water resources, by conducting cloud physics research and improving numerical model simulations of cloud seeding experiments. The K-CPEC began operation in 2021, along with the manufacture and installation of the aerosol and cloud chambers. This was followed by the test operation and performance evaluation of each chamber in 2022 (NIMS, 2022). In 2023, particle observation equipment was introduced

and chamber experiment procedures were developed. Various experiments on warm and cold clouds have been conducted at the K-CPEC since 2024 (NIMS, 2023). A substantial number of studies have used the cloud chamber for conducting homogeneous and heterogeneous nucleation for ice crystal formation and exploring the formation of ice crystals by secondary

aerosols. However, the studies that analyze the characteristics of materials for cloud seeding of warm clouds are rare. In this study, we analyzed the particle characteristics and cloud droplet growth of powder-type NaCl and $CaCl_2$ used for cloud seeding

in warm clouds using the K-CPEC. Section 2 presents the specifications of the K-CPEC setup, Section 3 presents the experimental and observation methods, Section 4 presents the results of the experiments, and Section 5 presents the major conclusion of this study.

## 2 Structure and features of the Korea Cloud Physics Experimental Chamber (K-CPEC)

### 2.1 Structure and features

The K-CPEC includes a cloud chamber and an aerosol chamber, as shown in Fig. 1; the functional specifications of each chamber are presented in Table 1. The cloud chamber, which has a double-structure comprising an outer and inner chamber, can be used for conducting experiments on droplet growth and ice-crystal formation by aerosol, by controlling the air pressure, wall temperature of the inner chamber, water-vapor content, and aerosol concentration (Cha et al., 2024). In the case of an expansion-type chamber, rapidly evacuating the air from the chamber lowers the air pressure and causing the air to cool

adiabatically. However, once the air becomes cooler than the chamber walls, the positive flux from the walls slows the cooling of the air by the adiabatic process (Wagner et al., 2020). To compensate for the heat, the walls of the inner chamber are cooled by a coolant. Therefore, the air temperature inside the inner chamber may decrease due to a quasi-adiabatic expansion process that simultaneously lowers the air pressure and wall temperature, as observed in the case of the double-structure chamber at the MRI (Tajiri et al., 2013). This structure is similar to the MRI chamber, but the inner-chamber volume of the K-CPEC is

22.4 $m^3$, which is 16-times larger than the chamber volume at the MRI.

The outer chamber is made of 304 grade stainless steel (thickness of 22 mm). Insulation (30 mm thick) is attached to the inside wall of the outer chamber; therefore, heat exchange between the cloud chamber and the inside of the K-CPEC facility is minimized. The inner chamber is made from stainless steel (4 mm thick), and the inside wall of the inner chamber has a meandering pattern with copper pipes (29 mm in diameter) passing from the left to right (at intervals of 40 mm) through each

panel of the octagonal prism structure (21 segments in total: 2 for the floor, 2 for the ceiling, 9 for the lower walls—including 2 segments forming the front door—and 8 for the upper walls). A thick copper plate (2 mm) was installed between each copper pipe. The copper pipes lowered the temperature of the wall surface comprising the copper pipes and plates as Novec 7200 heat transfer fluid cooled by the R507 HFC refrigerant of the cooling system flows. The heat transfer fluid was continuously circulated at a flow rate of 55 $m^3$ $h^{-1}$ through the copper pipes of all the walls of the inner chamber (with an approximate

residence time of 7 s per wall segment) and the stainless-steel pipes located between the cooling system and the cloud chamber (with an approximate circulation time of 98 s), using the brine supply pump of the cooling system. All the stainless-steel pipes along the refrigerant path are covered with insulation (100 mm thick) to minimize heat loss.

The inner chamber was not completely sealed, allowing air to be evacuated or supplied through gaps, primarily located on the ceiling, which were created for the installation of thermocouples and measurement instruments. The flow rate of the

vacuum pump of the cloud chamber was 1300 m³ h⁻¹ at 1800 RPM (maximum). A solenoid valve (SV) was installed at the top of the cloud chamber to control the flow rate of the vacuum pump by adjusting the opening rate. This flow rate control can generate an updraft velocity ranging from 0.1 to 19 m s⁻¹ inside the cloud chamber, based on an initial pressure of 1000 hPa. The dry air system included a triple filter, capable of removing particles of 10 μm, 5 μm, and 1 μm or larger, and a compressed air filter, capable of removing particles of 0.01 μm or larger, installed in the pipeline to supply clean, dry air to the cloud

chamber. Ultra-pure water with a conductivity of less than 0.055 μS cm⁻¹, produced by the pure water system, was supplied to the chamber, along with dry air passed through a Nafion tube, to control the RH inside the inner chamber; furthermore, water vapor was supplied at a flow rate of 300 L min⁻¹. We used a mixing fan to ensure that the air temperature, water vapor, and aerosol were spatially homogeneous throughout the inner chamber (Vallon et al., 2022).

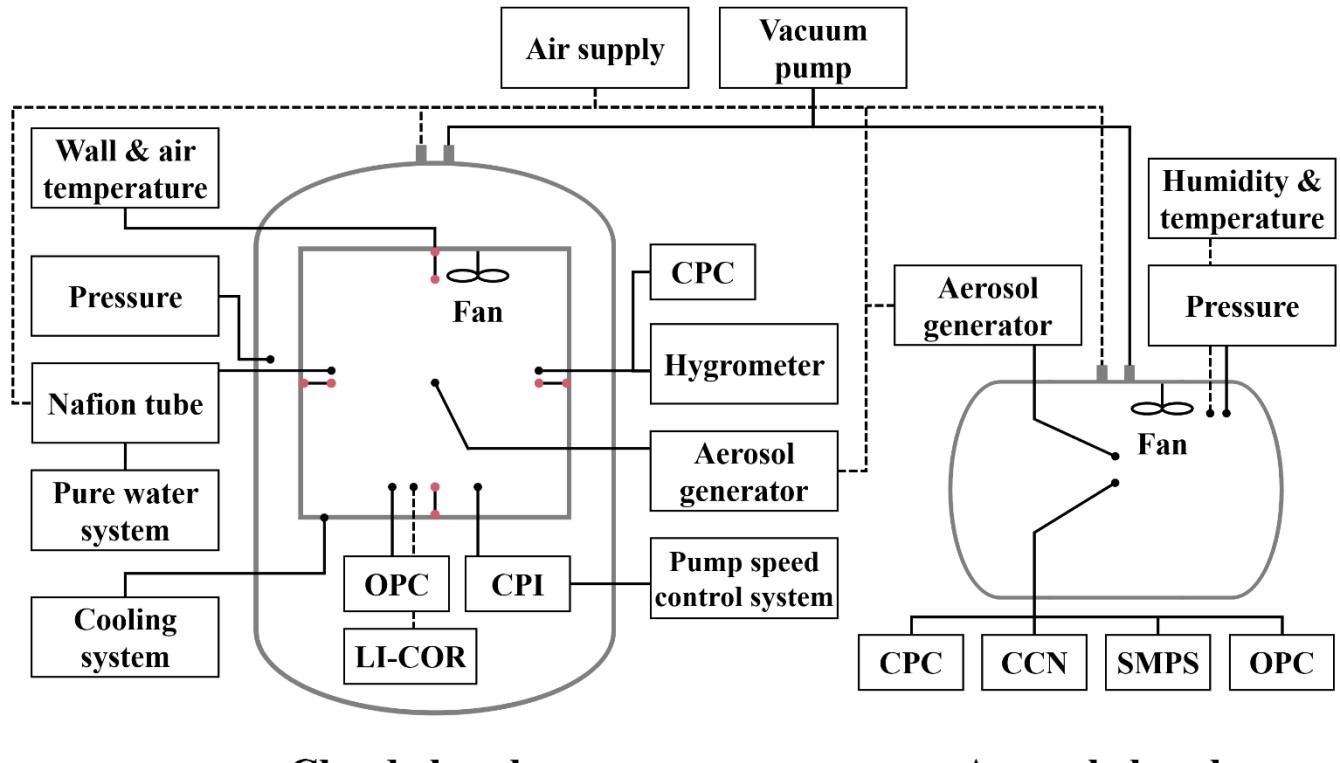

**Cloud chamber**              **Aerosol chamber**


**Figure 1. Schematic diagram of the Korea Cloud Physics Experimental Chamber (K-CPEC) and the cloud physics instruments used in this study, including a condensation particle counter (CPC), an optical particle counter (OPC), a cloud particle imager (CPI), a CO₂/H₂O analyzer (LI-COR), a cloud condensation nuclei counter (CCN), and a scanning mobility particle sizer (SMPS). Circles at the ends of the lines indicate action points for drawing air, injecting air, or conducting observations. The measurement locations of**

**the wall and air temperature in the cloud chamber are indicated by red points.**

**Table 1. Specifications of the cloud and aerosol chambers in the Korea Cloud Physics Experimental Chamber (K-CPEC) facility.**

| Type | | Shape / Size | Material / Thickness | Volume | Pressure / Temperature |
|---|---|---|---|---|---|
| Cloud chamber | Outer | Cylinder with elliptical ends / 7.5 m × 5 m | Stainless steel / 22 mm | 130 m³ | 1013.25–30 hPa (≤ ±0.3 hPa) / −70–60 °C (≤ ±0.5 °C) |
| | Inner | Octagonal prism / 3 m × 3 m | Stainless steel / 4 mm Copper / 2 mm | 22.4 m³ | |
| Aerosol chamber | | Cylinder with elliptical ends / 4.5 m × 3 m | Stainless steel / 14 mm | 28.3 m³ | 1013.25–30 hPa (≤ ±0.3 hPa) / ambient air temperature |

## 2.2 Instrumentation

The measurement instruments installed in the cloud chamber to observe aerosol particles and cloud droplets are listed in Table 2, and their measurement principles are summarized in Table 3. During the experiments, the wall and air temperatures (10 cm from the wall) of the inner chamber were measured using fast-response T-type thermocouples installed at the central points of the top, bottom, left, and right walls of the inner chamber (Dias et al., 2017). The relative humidity (RH) inside the inner chamber was calculated using the dew point temperature measured by a chilled mirror hygrometer (Buck Research

1011C), which drew air from the inner chamber for measurement, and the air temperature measured by the thermocouple (Buck Research Instruments, 2009). The aerosol was injected into the inner chamber at a constant rate using a rotating brush-type aerosol generator (Palas RBG1000). A condensation particle counter (CPC) was used to measure the total number concentration of condensation nuclei (CN) greater than 3 nm in size (in the injected aerosol). An optical particle counter (OPC) and a cloud particle imager (CPI) were used to measure both particle and droplet particles. The OPC measures particles in the

0.3–17 μm size range, whereas the CPI detects particles ranging from 10 μm to 2 mm using a high-speed camera (Connolly et al., 2007). Concerning the CPI, the area of the particle captured by the camera was used to calculate the equivalent diameter. In order to minimize the particle loss of hydrometeors in still air, the OPC and CPI were installed underneath the inner chamber with inlets oriented vertically upwards. As the cloud chamber experiment entailed a rapid decrease in pressure, a mass flow controller (MFC) was used to maintain a constant flow rate. In the case of the CPI, the constant mass flow was preserved using

a pump speed control system. Note that all the measurement instruments were connected to an external pump, to maintain a constant flow rate. All measurement instruments, except for the CPC and $CO_2$/$H_2O$ analyzer, sampled the air from the inner chamber and then vented it to the outer chamber after measurement. The LI-COR $CO_2$/$H_2O$ analyzer is an open-path instrument that measures the infrared absorption of water vapor in the air along the optical path between the transmitter and receiver, enabling the calculation of absolute humidity.

Unlike the cloud chamber, the aerosol chamber could not control the temperature directly. However, because dry air was supplied during the chamber-cleaning process, the aerosol chamber was slightly cooler than the ambient air and extremely dry (RH < 1 %), with the temperature typically at 20 °C ±5 °C. This enabled us to measure properties the aerosol particles at

a dry state. As in the cloud chamber, the aerosols were suspended using a mixing fan to ensure spatial homogeneity. The total

number concentration and particle size distribution (PSD) of aerosol particles with sizes of 11 nm–17 μm were measured using

an OPC and scanning mobility particle sizer (SMPS), respectively (Table 2). A cloud condensation nuclei (CCN) counter was

used to measure the concentration of the particles that were activated (condensed) under supersaturation conditions. In this

study, the CCNs were measured by dividing the supersaturation range (0.1–1 %) into intervals of 0.1 % while considering the

typical atmospheric supersaturation level (Loftus and Cotton, 2014). To ensure the stability of the CCN measurements, the

data of only the last 3 min for each interval were used. Since no experimental calibration of the CCN counter was conducted

in this study, the supersaturation values at each interval may carry an uncertainty of up to ±10 % (Rose et al., 2008). The air

temperature and RH in the aerosol chamber were measured using the Vaisala HMM170 sensor.

**Table 2. Specifications of the measurement instruments mounted in the cloud and aerosol chambers of the Korea Cloud Physics Experimental Chamber (K-CPEC) facility.**

| Type | Instrument | Flow setting | Observation range | Accuracy |
|---|---|---|---|---|
| Cloud chamber | Condensation particle counter (CPC, TSI 3750) | 1 L min$^{-1}$ | 7 nm (min.)– > 3 μm (max.) | ±5 % (< 10$^5$ particles) |
| | Optical particle counter (OPC, Walas Promo 2300) | 5 L min$^{-1}$ | 0.3–17 μm | Max. concentration for 10 % coincidence error (< 8×10$^3$ particles cm$^{-3}$) |
| | Cloud particle imager (CPI, TSI V2.5) | 5 L min$^{-1}$ | 10 μm–2 mm | ±2.3 μm for particle diameter (based on pixel resolution) |
| | Hygrometer (Buck Research 1011C) | 2.5 L min$^{-1}$ | –75–50 °C | ±0.1 °C |
| | CO$_2$/H$_2$O analyzer (LI-COR 7500DS) | - | 0–60 mmol mol$^{-1}$ | ±1 % for H$_2$O measurement |
| | Thermocouple (MSCT T type) | - | –270–400 °C | ±0.5 °C |
| | Pressure (Prignitz SPT-I2) | - | 0–1000 hPa | ±0.5 % |
| Aerosol chamber | Condensation particle counter (CPC, TSI 3750) | 1 L min$^{-1}$ | 7 nm (min.)– > 3 μm (max.) | ±5 % (< 10$^5$ particles) |
| | Cloud condensation nuclei counter (CCN counter, DMT CCN-200) | 0.5 L min$^{-1}$ | Size: 0.75–10 μm, Supersaturation: 0.1–1 % | Max. concentration for 10 % coincidence error (6×10$^3$ particles s$^{-1}$ below 0.2 %, 2×10$^3$ particles s$^{-1}$ above 0.3 %) |
| | Scanning mobility particle sizer (SMPS, TSI 3082) | 5 L min$^{-1}$ | 11–478 nm | ±1 % (< 10$^7$ particles cm$^{-3}$) |
| | Optical particle counter (OPC, Walas Promo 2070) | 5 L min$^{-1}$ | 0.3–17 μm | Max. concentration for 10 % coincidence error (< 8×10$^3$ particles cm$^{-3}$) |

| | Humidity & temperature (Vaisala HMM170) | - | 0–100 %, −70–180 °C | ±1 %, ±0.2 °C |
| | Pressure (Prignitz SPT-I2) | - | 0–1000 hPa | ±0.5 % |


**Table 3. Summary of measurement principles for the K-CPEC instruments.**

| Instrument | Measurement principle |
|---|---|
| CPC | Detects ultrafine particles by enlarging them through the condensation of a working fluid (typically butanol), making them optically countable. |
| SMPS | Classifies particles based on electrical mobility using a differential mobility analyzer (DMA) and measures their size distribution with a CPC. |
| OPC | Measures particle size and number concentration by analyzing light scattered by individual particles as they pass through a laser beam. |
| CCN counter | Measures the concentration of cloud condensation nuclei by exposing aerosol particles to a controlled supersaturation and counting those that activate into cloud droplets. |
| CPI | Captures high-resolution images of cloud particles using pulsed laser illumination and a CCD camera to analyze their size, shape, and phase. |
| LI-COR | Uses non-dispersive infrared spectroscopy to measure water vapor and $CO_2$ concentrations based on their absorption of specific infrared wavelengths. |
| Thermocouple | Measures temperature via thermoelectric voltage generated between copper and constantan, offering fast response and high accuracy at low temperatures. |
| Hygrometer | Determines dew point temperature by cooling a mirror until water vapor condenses on its surface. |

## 3 Experimental procedure and observation methods

In this study, powder-type NaCl and $CaCl_2$ materials with hygroscopic properties, as shown in Table 4, were used. Both the
materials were milled to ensure a wide range of size distribution (from nm to μm), using the air jet milling method; the materials were milled from raw materials with a purity of 96 % or higher. Air jet milling produces fine powders by inducing particle–particle collisions through high-velocity compressed air (Kou et al., 2017). Ultra-giant CCNs with sizes of 10 μm or larger can promote early precipitation formation in warm clouds (Segal et al., 2004). Giant CCNs with sizes of 1–10 μm and large CCNs with size < 1 μm can expand the cloud droplet size distribution (DSD) and accelerate the formation of large droplets by
promoting the collision-coalescence process (Bruintjes, 1999; Silverman, 2003). In addition, Aitken nuclei can also act as

CCNs in high-supersaturation conditions. Therefore, cloud seeding materials that can broaden the size spectrum of cloud droplets in clouds are more efficient in forming large droplets (Wang et al., 2024). NaCl and $CaCl_2$ have different hygroscopicity parameters ($\kappa$ = 1.24 and 0.78, respectively) and deliquescence relative humidities (DRH = 75 % and 28 %, respectively) (Fountoukis and Nenes, 2007; Liu et al., 2014). Consequently, they are expected to exhibit different deliquescence transitions and hygroscopic growth behaviors during cloud chamber experiments.

In the aerosol chamber, the PSD of each material was measured using the SMPS and OPC; the activation fraction ($F_{act}$) and activation diameter ($D_{act}$) of the CCNs were calculated using the CPC and CCN counter, according to the supersaturation level. Note that $F_{act}$ denotes the fraction (%) of CN ($N_{CN}$) of the CPC and the CCN ($N_{CCN}$) of the CCN counter according to the supersaturation, calculated using Eq. (1). Furthermore, $D_{act}$ refers to the diameter at which the fraction of the cumulative number concentration (from large to small sizes) equals the $F_{act}$, assuming that all the particles in the size-resolved number concentration distribution measured by the SMPS and OPC are activated (refer to Eq. (2)) (Hung et al., 2014). In other words, $D_{act}$ represents the diameter of the smallest particle that can activated at the corresponding supersaturation.

$$F_{act} = \frac{N_{CCN}}{N_{CN}} \tag{1}$$

$$\frac{\int_{D_{act}}^{max} dNdlogD}{\int_{min}^{max} dNdlogD} = \frac{N_{CCN}}{N_{CN}} \tag{2}$$

In the cloud chamber experiment, for the SV values (of the vacuum pump) of 20 % and 50 %, the experiments lasted 900 and 840 s, respectively. The air in the cloud chamber was evacuated through a pipe (with diameter of 200 mm). Although the design allowed for the simultaneous use of two vacuum pumps, in this study, we used only one vacuum pump; the SV value of 50 % resulted in the same evacuation rate as the SV value of 100 % owing to the large diameter of the evacuation pipe. Note that the air temperature ($T_{air}$) and wall temperature ($T_{wall}$) were expressed as the mean of the measurements conducted at four points in the inner chamber. The mean standard deviations of both the $T_{air}$ and $T_{wall}$ were ±0.2 °C, depicting a similar temperature distribution. The $T_{air}$ and dew point temperature ($T_{dew}$) measured using the hygrometer were used to calculate the RH, expressed as $RH_w$ and $RH_i$ for the water and ice phases, respectively. During the experiment in the cloud chamber, the air pressure and temperature decreased, and supersaturation conditions exceeding 100 % RH were created in the inner chamber via a quasi-adiabatic expansion process. The cloud droplets formed during this process were observed using the OPC and CPI. The cloud DSD was constructed at 1 s intervals, and mean diameter was calculated by averaging the DSDs over the observation period using both datasets.

**Table 4. Specifications of the NaCl and CaCl₂ powders used in this study.**

| Aerosol type | Purity (wt. %) | Other content (wt. %) |
|---|---|---|
| NaCl (CAS No. 7647-14-5) | 96 % (min.)–99.56 % (test) | $H_2O$ 0.05 %, unknown (As, Cd, Pb were not detected) |
| $CaCl_2$ (CAS No. 10043-52-4) | 96 % (min.)–98.28 % (test) | $Ca(OH)_2$ 1.5 %, unknown (As, Cd, Pb were not detected) |

The K-CPEC facility maintains an indoor air temperature of 24 °C and RH of less than 60 % to comply with the operating specifications of the measurement instruments and provide a consistent experimental environment. In this study, the experiments using the same method or material were conducted on the same day under similar weather conditions to ensure the most consistent experimental initial conditions for each trial (Cheng et al., 2024). The aerosol chamber experiment was conducted using the following process: Step 1: Turn on the main control PC and measurement instruments; Step 2: Clean the chamber; Step 3: Set the reference air pressure in the aerosol chamber; Step 4: Inject the aerosol into the aerosol chamber using an aerosol generator; Step 5: Begin the experiment and measurement; and Step 6: Clean the chamber. The operation of the main control PC and measurement equipment involved the time-synchronization of the system. The cleaning of the aerosol chamber was repeated 1–2 times between the ambient air pressure and 30 hPa. When the air pressure in the aerosol chamber reached 30 hPa, dry air was supplied to the chamber; the chamber was flushed with dry air for 5 to 10 min. During the cleaning process, the mixing fan was set to a 1200 RPM (maximum) to suspend the remaining aerosol in the aerosol chamber, to perform cleaning. If the remaining number concentration of the aerosol was more than 10 $cm^{-3}$, the cleaning procedure was performed once more. The reference pressure for the aerosol chamber experiment was set to 30 hPa, lower than the ambient air pressure, to facilitate aerosol injection. The NaCl and $CaCl_2$ powders were injected into the aerosol chamber at a rate of 80 mm $h^{-1}$, using a brush-type aerosol generator. The experiment in the aerosol chamber was conducted for approximately 1 h, depending on the supersaturation interval setting of the CCN counter (for the range 0.1–1 %, with intervals of 0.1 %; 10 min of observation for the 0.1 % interval, and 5 min for the remaining intervals). The SMPS and OPC data measured during the observation period (of 1 h) were mean to each size bin, to calculate the PSD. The DMA of the SMPS determines particle size based on electrical mobility, whereas the OPC measures the optical diameter, which depends on the refractive index (RI) of the particle. Accordingly, the size parameter of OPC was set based on the RI of 1.54 for NaCl and $CaCl_2$ (Zinke et al., 2022). In addition, the ratio of the sheath and sample flows of the SMPS was set to 10:1. During the experiment, the mixing fan was set to 300 RPM, to ensure a spatially homogeneous distribution of the aerosol within the chamber. The environmental conditions of the aerosol chamber experiment performed in this study are shown in Table 5. During both experiments, the temperature, pressure, and relative humidity in the aerosol chamber were controlled, with mean standard deviations of 0.05 °C, 4.73 hPa, and 0.08 %, respectively.

The cloud chamber experiment was conducted as follows: Step 1: Turn on the main control PC and measurement instruments; Step 2: Clean the chamber; Step 3: Set the reference air pressure and wall temperature in the cloud chamber; Step 4: Supply water-vapor to the chamber to achieve the reference RH; Step 5: Inject the aerosol into the cloud chamber using an

aerosol generator; Step 6: Begin the experiment and observation (with vacuum pump operation and wall temperature adjustment); and Step 7: Clean the chamber. The cleaning process was repeated 1–2 times, with the air pressure cycled between ambient pressure and 150 hPa to prevent damage to the instruments. The chamber was flushed dry air and the mixing fan settings were the same as those applied in the aerosol chamber experiment. The reference pressure of the cloud chamber was set to 100 hPa lower than the ambient air pressure, considering the air pressure increase caused by the supply of water vapor and aerosols. The wall and air temperatures were set to approximately 20 ℃. Then, ultra-pure water produced by the pure water system, along with dry air, was passed through a Nafion tube to generate water vapor. Then, the water vapor was supplied to the inner chamber of the cloud chamber, to adjust the RH (< 60 %). In this study, the formation of cloud droplets was simulated by injecting aerosols that were representative of polluted air levels (Hudson and Noble, 2014; Grabowski et al., 2022). The aerosol was injected into the chamber at a rate of 80 mm h$^{-1}$. In the cloud chamber experiment, the particles measured by the OPC were assumed to be water droplets. Therefore, the size parameter of the OPC was set based on the RI value of 1.33 (water). During the experiment, the mixing fan was set to 300 RPM, to ensure spatially homogeneous distributions of the air temperature and aerosol concentration within the inner chamber and facilitate the suspension of particles and droplets (Vallon et al., 2022). The environmental conditions of the cloud chamber experiment performed in this study are shown in Table 6.

**Table 5. Experimental conditions in the aerosol chamber for NaCl and CaCl$_2$ powders. The values in parentheses indicate the standard deviations of environmental conditions.**

| Type | Air temperature (℃) | Air pressure (hPa) | Relative humidity (%) | Total number concentration (cm$^{-3}$) |
|---|---|---|---|---|
| NaCl | 22.97 (0.05) | 940.79 (4.64) | 0.05 (0.07) | 1185.34 |
| CaCl$_2$ | 22.95 (0.05) | 939.64 (4.82) | 0.13 (0.07) | 1142.65 |

**Table 6. Experimental conditions in the cloud chamber for NaCl and CaCl$_2$ powders (N: total number concentration, $T_{air}$: air temperature, $T_{wall}$: wall temperature, $T_{dew}$: dew point temperature, P: air pressure, RH: relative humidity, and SV: solenoid valve).**

| Exp. Name | Type | N (cm$^{-3}$) | $T_{air}$ (℃) | $T_{wall}$ (℃) | $T_{dew}$ (℃) | P (hPa) | RH (%) | SV (%) |
|---|---|---|---|---|---|---|---|---|
| NaCl Exp. #1 | NaCl | 1052.31 | 19.91 | 20.32 | 10.41 | 981.81 | 54.87 | 20 |
| NaCl Exp. #2 | NaCl | 1128.55 | 19.72 | 20.10 | 10.48 | 982.75 | 55.76 | 50 |
| CaCl$_2$ Exp. #1 | CaCl$_2$ | 1040.36 | 20.11 | 20.18 | 11.24 | 953.12 | 57.27 | 20 |

| CaCl₂ Exp. #2 | CaCl₂ | 1071.92 | 19.23 | 19.20 | 10.84 | 949.69 | 58.91 | 50 |
|---|---|---|---|---|---|---|---|---|

## 4 Results

### 4.1 Aerosol chamber experiments on the characteristics of NaCl and CaCl₂ powders

The PSD of the NaCl and CaCl₂ powders used in this study are shown in Fig. 2. These distributions were merged with those
of OPC values measured after the maximum size measurable by SMPS, that is, the PSDs of 11 nm–0.48 μm were measured
using SMPS and the PSDs of 0.48–17 μm were measured using OPC, with each distribution representing 1 h mean values for
each size bin. Notably, the data in Fig. 2 were obtained simultaneously during the same experiment as those shown in Fig. 3
for the CPC and CCN counter measurements. The PSDs of both powders exhibited bimodal distribution, and the number
concentration peaks in the submicron size were 0.19 μm and 0.24 μm (for NaCl and CaCl₂, respectively); the number
concentration peaks in the micron size for both the powders were at 1.84 μm. The mean particle size of NaCl was 0.37 μm,
smaller than that of CaCl₂ (0.44 μm). The particles with diameters $\geq$ 1 μm accounted for 8.75 % and 12.31 % of the total
number concentrations of NaCl and CaCl₂, respectively; the particles with diameters $\geq$ 10 μm accounted for 0.19 % and 0.12 %,
respectively. In aerosol chamber experiments, high aerosol concentrations combined with prolonged residence times may
promote particle coagulation (Chen et al., 2023). However, in this study, the total number concentration during both
experiments ranged from 1100 to 1200 cm$^{-3}$, suggesting that the effect of coagulation on the PSD was minimal. Thus, the sizes
of the majority of particles were smaller than 1 μm, but the particles were characterized by a wide range of PSDs. In contrast
to this result, the submicron size peak of the burn-in-place type hygroscopic flare, which is widely used for cloud seeding
worldwide, was very small (0.1 μm) (Kuo et al., 2024, Miller et al., 2024). As the flare particles were much smaller in size
compared to the typical powder-type materials, their activation as CCNs in low-supersaturation conditions was limited;
therefore, an immediate seeding effect was difficult to observe in the cloud seeding experiment (Dong et al., 2023).

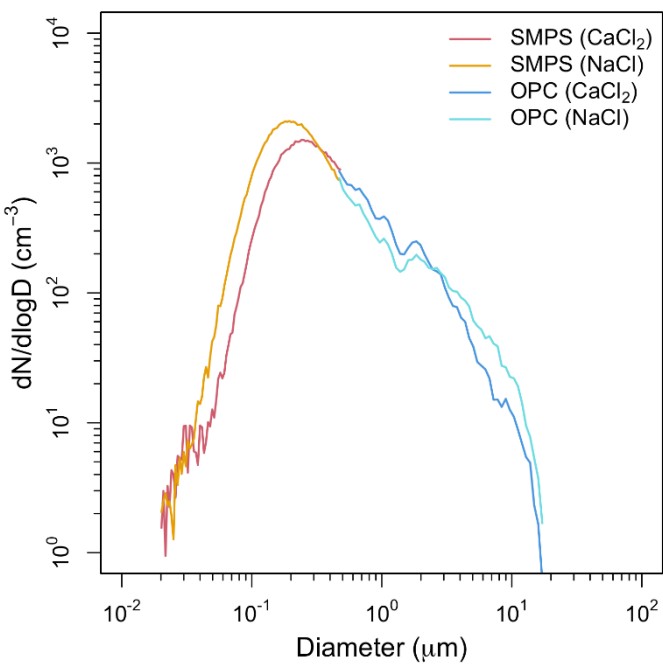

**Figure 2. Distribution of size-resolved number concentration of NaCl and CaCl₂ powders based on the scanning mobility particle size (SMPS) and optical particle counter (OPC) observations.**


The $F_{act}$ and $D_{act}$ of NaCl and CaCl₂ powders according to the supersaturation level (S) are shown in Fig. 3. The red points represent the CCN data for each supersaturation interval (with the range being 0.1–1 %, at intervals of 0.1 %) considered in this study. The $F_{act}$ and $D_{act}$ were calculated using the CN data acquired using the CPC, measured simultaneously with the CCN data. In the case of CaCl₂ powder, the $F_{act}$ was 92.30 % at S of 0.1 %, while the NaCl powder exhibited a relatively low
$F_{act}$ of 79.33 %. This may be because, as shown in Fig. 2, the NaCl powder contained numerous particles that were not large enough to act as CCN at S of 0.1 %. When S was 0.1 %, the $D_{act}$ of the NaCl powder was 135.8 nm and that of CaCl₂ powder was 126.3 nm, with a particle size difference of approximately 10 nm. Notably, hygroscopic materials with relatively large particles can have more particles acting as CCNs, even in low supersaturation conditions (Rose et al., 2010). In this study, since $D_{act}$ was determined using the method proposed by Hung et al. (2014), it may differ from the critical diameter ($D_{crit}$)
measured using a DMA (e.g., $D_{crit}$ of NaCl = 100 ± 4 nm at S = 0.1 %; Niedermeier et al., 2008). For the NaCl powder, the $F_{act}$ value at S = 0.2 % was greater than 90 %, and both the powders exhibited $F_{act}$ values greater than 95 % at S of 0.3 %. As the value of S increased, the $F_{act}$ values of both powders increased to almost 100 %, and the $D_{act}$ values were approximately 70 nm, indicating that the particles could act as CCNs up to size of Aitken mode. Note that for S ≥ 0.4 %, the CaCl₂ powder exhibited a consistent $F_{act}$ value. Thus, the CaCl₂ powder may be suitable for cloud seeding experiments for relatively low
supersaturation conditions. The NaCl is considered to be suitable for cloud seeding experiments in conditions of high

supersaturation, i.e., near cumulus clouds with strong updrafts or in the vicinity of mountain ranges with strong orographic updrafts (Cotton et al., 2007; Li et al., 2023).

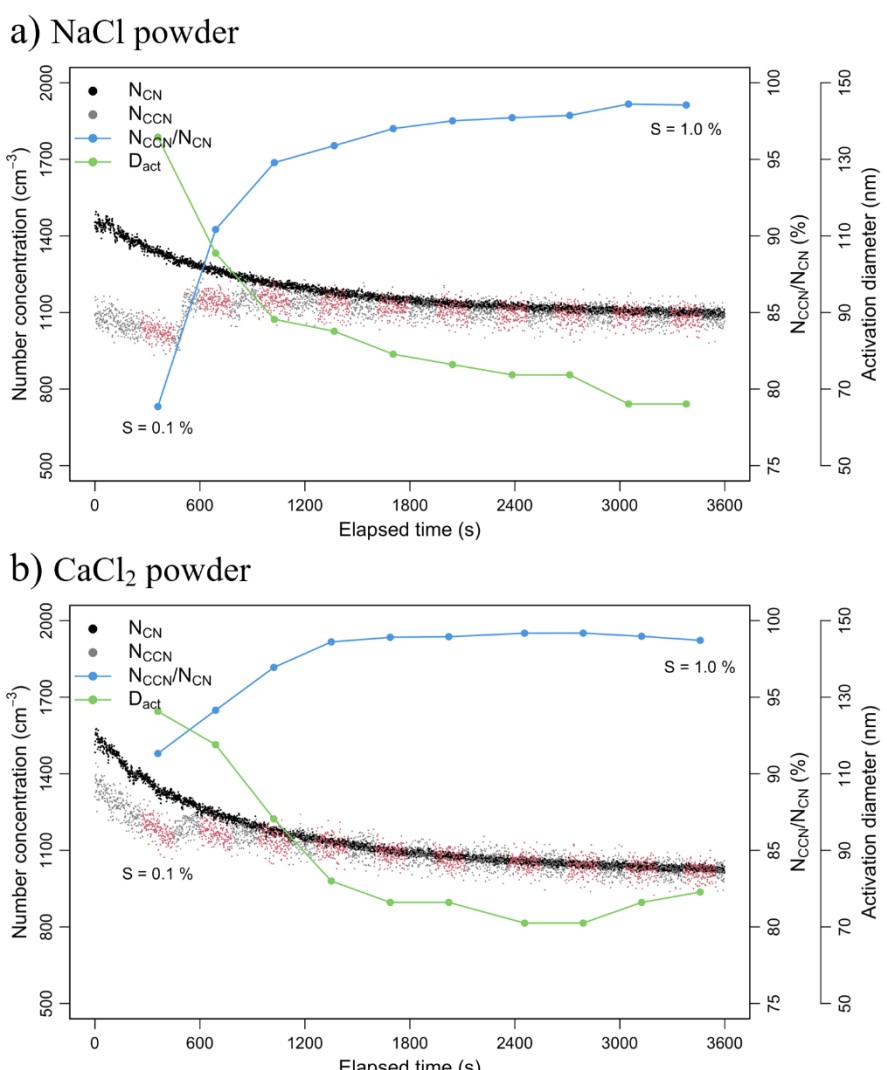

**Figure 3. Total number concentration of condensation nuclei (CN) measured using CPC ($N_{CN}$) and cloud condensation nuclei (CCN) concentration measured using a CCN counter ($N_{CCN}$), along with activation fraction ($N_{CCN}/N_{CN}$) and activation diameter ($D_{act}$) for (a) NaCl and (b) CaCl$_2$ powders across all the supersaturation (S) intervals (0.1–1 %, with intervals of 0.1 %), based on 3-min data collected at 1-s intervals (depicted using red points).**

## 4.2 Cloud chamber experiments on the observations of cloud droplet formation

The results of the cloud chamber experiments conducted using NaCl (NaCl Exp. #1 and #2) and $CaCl_2$ (NaCl Exp. #1 and #2) are shown in Figs. 4 and 5, respectively. The air pressure, $T_{air}$, $T_{dew}$, $T_{wall}$, RH, updraft velocity, cooling rate, lapse rate, absolute humidity, size-dependent number concentration ($D > 0.3$, $> 1$, $> 5$, $> 10$, and $> 20$ μm), size-resolved number concentration, and mean diameter of droplets in the inner chamber are shown in the figures. The maximum and mean values of the updraft velocity and cooling rate of the cloud chamber for each experiment are shown in Table 7. In each experiment, the time required for the SV rate to increase from 0 % to 20 % and 50 % was 6 and 10 s, respectively. The time required for the vacuum pump to reach an RPM of 1800 (from 0) was 30 s. In the NaCl Exp. #2, the maximum updraft velocity (18.3 m s$^{-1}$) could be achieved in 55 s after the beginning of the experiment; the mean updraft velocity was 13.5 m s$^{-1}$. The maximum cooling rate 85 s after the beginning of the experiment was –7.3 K min$^{-1}$; the mean cooling rate was –2.3 K min$^{-1}$. These experimental conditions may simulate cloud seeding experiments in areas with weak-to-strong convection and strong orographic updrafts (Jensen et al., 1998; Field et al., 2001). Within the first approximately 150 s of the cloud chamber experiment, the heat transfer fluid circulated through the inner chamber, rapidly lowering the wall temperature before stabilizing. A full circulation of the heat transfer fluid through the cloud chamber and the cooling system took approximately 98 s. This phenomenon occurred temporarily before the RH in the internal chamber reached 100 % and did not significantly affect the variation in air temperature.

In NaCl Exp. #1 and #2, the total number concentrations of the injected powder were 1052.31 cm$^{-3}$ and 1128.55 cm$^{-3}$, respectively (Table 6). Immediately after the beginning of the experiment, particles larger than 0.3 μm accounted for approximately 56 % of the total number concentration for an RH of approximately 55 %. This figure was 22 % higher than the proportion of the number concentration (approximately 34 %) for particles larger than 0.3 μm measured in the aerosol chamber experiment with NaCl powder. Initially, the RH remained below the DRH, and thus, the deliquescence transition was not activated. However, owing to the strong hygroscopicity of NaCl, water vapor might have been taken up even below the DRH, leading to pre-deliquescence hygroscopic growth characterized by the formation of a thin water layer (Tang and Munkelwitz, 1993). After RH exceeded 70 % (Exp. #1: 87 s; Exp. #2: 56 s), the deliquescence transition was initiated and continued until RH reached approximately 85 % (Exp. #1: 224 s; Exp. #2: 90 s) (Peng et al., 2022). During this period, particles smaller than 2.84 μm—corresponding to the intermodal size of the bimodal distribution—grew in size, and the number concentration increased by 13.73 % in Exp. #1 and 7.78 % in Exp. #2. In addition, the number concentration of the particles with PSDs of 2.84–17 μm decreased by 25.81 % in Exp. #1 and 3.61 % in Exp. #2, which may be related to large cloud droplets (several tens of μm in size, with the captured by the CPI). After RH exceeded 85 % (Exp. #1: 224 s; Exp. #2: 90 s), post-deliquescence hygroscopic growth continued. This period showed the most significant decrease in mean absolute humidity, with values of 0.012 g m$^{-3}$ per second in Exp. #1 and 0.016 g m$^{-3}$ per second in Exp. #2. When RH exceeded 100 % (Exp. #1: 430 s; Exp. #2: 316 s), cloud droplet formation occurred through condensational growth. During this phase, the mean lapse rate ($-dT/dz$) was 4.26 K km$^{-1}$ in Exp. #1 and 2.28 K km$^{-1}$ in Exp. #2. This finding indicates that Exp. #1 followed the wet adiabatic lapse rate ($\Gamma_w$, 3.5–6.5 K km$^{-1}$; Weiner and Matthews, 2003), whereas Exp. #2 exhibited a more stable environmental lapse rate,

lower than $\Gamma_w$. Notably, cloud droplets that can be formed in an environment where sufficient water-vapor is supplied in a clean atmosphere are typically 20–30 μm in size (Li et al., 2017). In comparison, the artificial injection of hygroscopic materials, such as NaCl, can facilitate the growth of initial cloud droplets. Under super-saturated (RH > 100 %) conditions, cloud droplets of 30–50 μm in size were consistently observed not only in NaCl Exp. #1 but also in the other three experiments. These cloud droplets may have formed earlier at the center of the inner chamber; these droplets may have required time to reach the bottom of the inner chamber, where the observations were conducted using OPC and CPI (Frey et al., 2018).

With respect to the both powders used for the cloud seeding experiment, the cloud DSDs (see Figs. 4 and 5 were not constant due to the fluctuations in supersaturation. In addition, as the air in the inner chamber was evacuated using a vacuum pump, the CCNs or droplets in the chamber may have been lost. Furthermore, condensation also occurred on the inner chamber walls, which could cause it to rapidly loss in supersaturated conditions (Shao et al., 2022). The observation results clearly indicated that the process through which the particles grew after the RH reached 100 % occurred under supersaturated conditions. The growth process of cloud droplets was divided into the under-saturated stage (RH ≤ 85 %, hereinafter referred to as S1), pre-saturated stage (85 % < RH ≤ 100 %, hereinafter referred to as S2), and super-saturated stage (RH > 100 %, hereinafter referred to as S3), as shown in Fig. 6, and the cloud DSD—possibly including both aerosols and droplets—was expressed. These stages were divided based on the variations in the cloud DSDs observed in different RH ranges; for example, in NaCl Exp. #1, the cloud DSD shown in Fig. 4i for 1–223 s exhibits a monomodal distribution with a peak at 0.58 μm, similar to the observations for S1 shown in Fig. 6a. However, once the RH exceeded 85 %, the monomodal distribution transitioned to a bimodal distribution (with a second mode peak at 7.78 μm), similar to the observations for S2 shown in Fig. 6a. This bimodal distribution persisted until RH exceeded 100 %; in this supersaturated stage, the right tail of the bimodal distribution increased relatively, as observed in the S3 shown in Fig. 6a. The maximum cloud droplet size ($D_{max}$) was 65.71 μm (observed at 832 s) in this experiment. In the case of NaCl Exp. #2, the SV opening rate was increased from 20 % to 50 %, resulting in the evacuation speed being nearly twice as fast as in NaCl Exp. #1. Therefore, the process in the S1 shown in Fig. 6b was shortened to 90 s, and the time required to reach 100 % RH was reduced to 316 s, reaching supersaturation more than 100 s faster compared to that observed in NaCl Exp. #1. Furthermore, in the S3 (Fig. 6b), the peak of the second mode was at 26.43 μm, and the $D_{max}$ was 89.16 μm (observed at 335 s). Although the number concentration inside the inner chamber decreased rapidly compared to that in the NaCl Exp. #1, large cloud droplets continued to form and were observed until the end of the experiment. This phenomenon indicated that as the supersaturation condition was maintained until the end of the experiment, the cloud particles at below freezing point were observed by the CPI as spherical supercooled water droplets. The maximum mean cloud droplet size ($D_{mean}$) in NaCl Exp. #1 and NaCl Exp. #2 was 27.87 and 40.71 μm, respectively (Figs. 4i and 4j). These large cloud droplets have a size distribution corresponding to the important diameter for droplets acting as drizzle embryos, leading to rain droplets (Zhu et al., 2024).

In CaCl$_2$ Exp. #1 and #2, CaCl$_2$ powder was injected into the inner chamber at total number concentrations of 1040.36 and 1071.92 cm$^{-3}$, respectively (Table 6). Immediately after the beginning of the experiment, the number concentration of particles larger than 0.3 μm was approximately 80 % compared with the injected total number concentration under an RH of

approximately 58 %. This figure is 27 % higher than the proportion of number concentration (approximately 53 %) for particles larger than 0.3 μm measured in the aerosol chamber experiment with $CaCl_2$ powder. Owing to the low DRH of $CaCl_2$ (28 %), a large number of particles might have undergone deliquescence transition immediately upon injecting $CaCl_2$ powder into the cloud chamber (Guo et al., 2019). Therefore, before the RH reached 85 %, the mean particle size within the size distribution below 2.84 μm was 1.06 μm, which was larger than the mean particle size of 0.94 μm in the NaCl experiments. In addition, the mean number concentration was approximately 692 $cm^{-3}$, which was 45 % higher than that in the NaCl experiments (Xueling et al., 2021; Peng et al., 2022). Therefore, in cloud seeding experiments conducted using an aircraft, $CaCl_2$ powder can rapidly increase the number concentration of potential CCNs in low RH conditions. The S1 and S2 stages in $CaCl_2$ Exp. #1 and #2 progressed faster than those in NaCl Exp. #1 and #2.

The hygroscopicity of NaCl is known to be higher than that of $CaCl_2$ (Kumar et al., 2011). Note that the stronger the hygroscopicity of a material, the more easily the CCNs can uptake the surrounding water vapor; this may cause a hygroscopic buffering effect that delays the increase in RH in the chamber (Ding et al., 2024). For the NaCl powder (with high hygroscopicity), the time required for the RH inside the inner chamber to reach 100 % was delayed by approximately 50 s. Thus, the process of cloud droplet growth differed depending on the deliquescence and hygroscopicity characteristics of the seeding material. The cloud DSDs in the S1–S3 stages in $CaCl_2$ Exp. #1 and #2 were similar to those observed in NaCl Exp. #1 and #2. The maximum $D_{mean}$ observed in $CaCl_2$ Exp. #1 was 24.64 μm, while that in $CaCl_2$ Exp. #2 was 38.30 μm, indicating a difference of 2–3 μm smaller compared with the mean diameters observed in the NaCl experiments. The $D_{max}$ of the droplets observed in $CaCl_2$ Exp. #1 and #2 were 54.99 (observed at 458 s) and 68.42 μm (observed at 548 s), respectively, 10–20 μm smaller than the $D_{max}$ values observed in NaCl Exp. #1 and #2. In this study, the cloud chamber experiments were performed under similar conditions, using NaCl and $CaCl_2$ powders. However, the difference in droplet growth indicated greater growth in NaCl powder, which exhibited relatively higher hygroscopicity. The mean $D_{max}$ values for S1–S3 stages in each experiment are summarized in Table 8. In the S1 stage, NaCl and $CaCl_2$ each showed a small difference (approximately 1 μm) between Exp. #1 and #2, with NaCl consistently exhibiting mean $D_{max}$ values that were 4–5 μm larger than those of $CaCl_2$. In the S2 stage, where hygroscopic growth was active under pre-saturated conditions, cloud droplets showed a larger mean $D_{max}$ by 15–20 μm compared with that at the S1 stage. Although the largest cloud droplets appeared during the S3 stage, when condensational growth was dominant, the mean $D_{max}$ was smaller than that at the S2 stage. This is because although the RH inside the cloud chamber was consistently exceeded 100 %, the absolute humidity decreased as the experiment progressed. In other words, these findings suggest that a larger mean $D_{max}$ can be observed in an environment where sufficient water vapor is supplied.

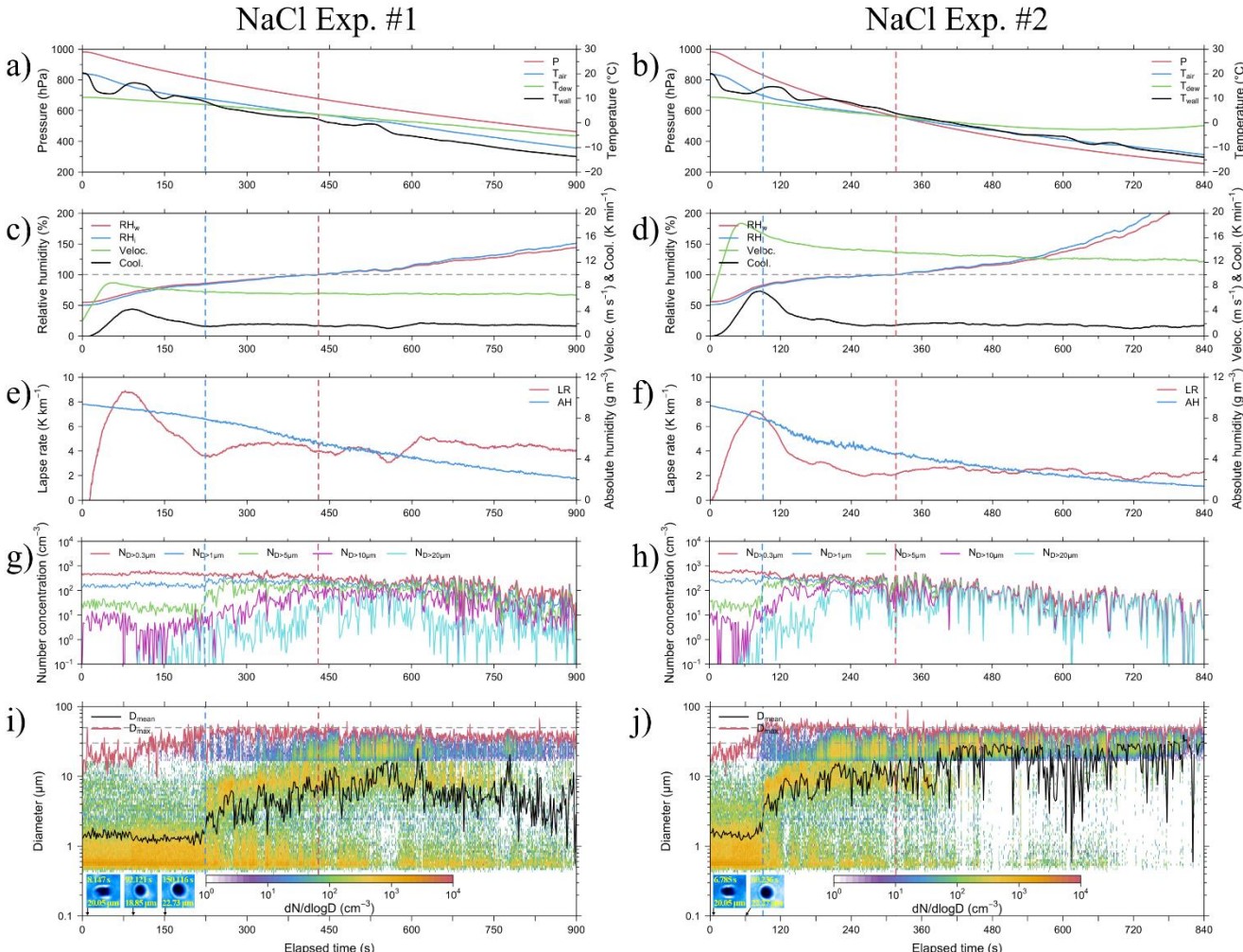

**Figure 4. Results from NaCl Exp. #1 with SV of 20 % (left) and NaCl Exp. #2 with SV of 50 % (right), presented in 1-s intervals. (a, b) present plots for air pressure (P, red line), with respect to air temperature ($T_{air}$, blue line), dew point temperature ($T_{dew}$, green line), and wall temperature ($T_{wall}$, black line). (c, d) presents plots for relative humidity for water ($RH_w$, red line), with respect to relative humidity for ice ($RH_i$, blue line), updraft velocity (Veloc., green line), and cooling rate (Cool., black line). (e, f) present the plots for lapse rate (LR, red line) and absolute humidity (AH, blue line). (g, h) present the plots for number concentration of droplets with diameters > 0.3 µm ($N_{D>0.3µm}$, red line), > 1 µm ($N_{D>1µm}$, blue line), > 5 µm ($N_{D>5µm}$, green line), > 10 µm ($N_{D>10µm}$, magenta line), and > 20 µm ($N_{D>20µm}$, cyan line). (i, j) presents the plots for size-resolved number concentration, mean diameter ($D_{mean}$, black line), and maximum diameter ($D_{max}$, red line), with the horizontal gray dashed lines indicating diameters of 30 and 50 µm, respectively, and the images present particles captured by the CPI under RH < 85 % conditions. The vertical blue and red dashed lines indicate $RH_w$ of 85 % and 100 %, respectively.**

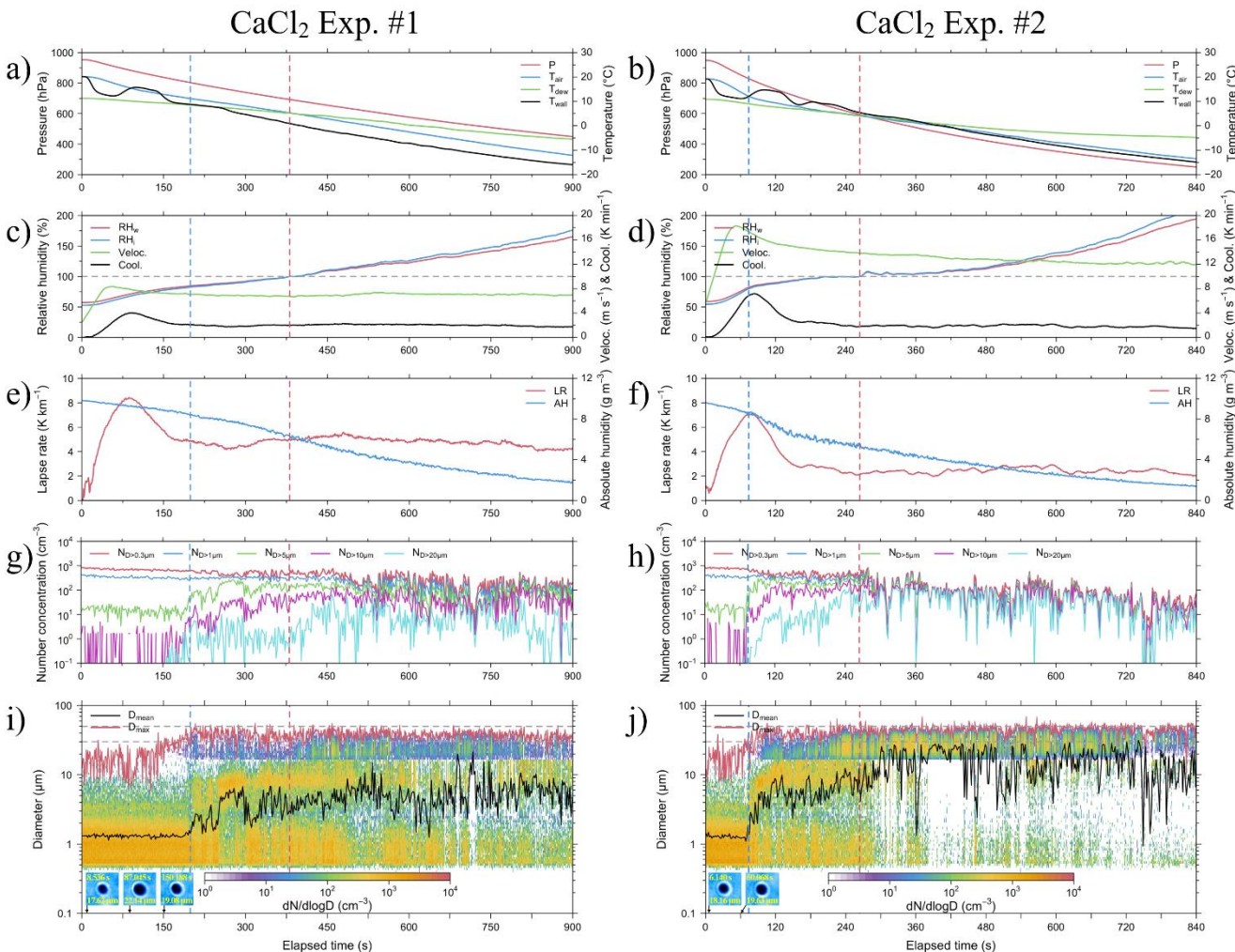

Figure 5. Results from CaCl₂ Exp. #1 with SV of 20 % (left) and CaCl₂ Exp. #2 with SV of 50 % (right), presented in 1-s intervals. (a, b) present plots for air pressure (P, red line), with respect to air temperature ($T_{air}$, blue line), dew point temperature ($T_{dew}$, green line), and wall temperature ($T_{wall}$, black line). (c, d) presents plots for relative humidity for water ($RH_w$, red line), with respect to relative humidity for ice ($RH_i$, blue line), updraft velocity (Veloc., green line), and cooling rate (Cool., black line). (e, f) present the plots for lapse rate (LR, red line) and absolute humidity (AH, blue line). (g, h) present the plots for number concentration of droplets with diameters > 0.3 µm ($N_{D>0.3µm}$, red line), > 1 µm ($N_{D>1µm}$, blue line), > 5 µm ($N_{D>5µm}$, green line), > 10 µm ($N_{D>10µm}$, magenta line), and > 20 µm ($N_{D>20µm}$, cyan line). (i, j) presents the plots for size-resolved number concentration, mean diameter ($D_{mean}$, black line), and maximum diameter ($D_{max}$, red line), with the horizontal gray dashed lines indicating diameters of 30 and 50 µm, respectively, and the images present particles captured by the CPI under RH < 85 % conditions. The vertical blue and red dashed lines indicate $RH_w$ of 85 % and 100 %, respectively.

**Table 7. Maximum and mean values of updraft velocity and cooling rate for NaCl and CaCl₂ experiments.**

| Variable | Condition | NaCl Exp. #1 | NaCl Exp. #2 | CaCl$_2$ Exp. #1 | CaCl$_2$ Exp. #2 |
|---|---|---|---|---|---|
| Updraft velocity (m s$^{-1}$) | Max | 8.69 | 18.35 | 8.37 | 18.30 |
| | Mean | 7.02 | 13.52 | 7.03 | 13.37 |
| Cooling rate (K min$^{-1}$) | Max | –4.39 | –7.31 | –4.03 | –7.18 |
| | Mean | –1.95 | –2.27 | –2.09 | –2.29 |

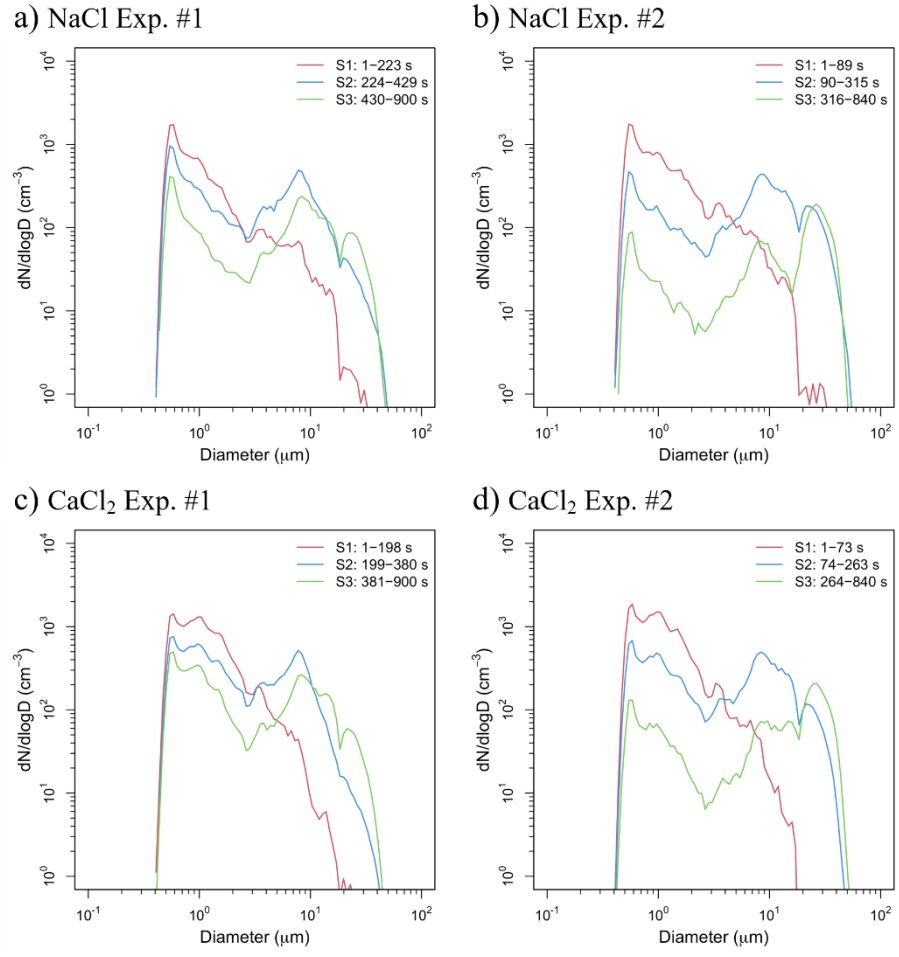

**Figure 6. Size-resolved number concentrations for the experiments conducted using (a, b) NaCl and (c, d) CaCl₂. S1 represents the under-saturated stage (RH ≤ 85 %, red line), S2 the pre-saturated stage (85 % < RH ≤ 100 %, blue line), and S3 the supersaturated stage (RH > 100 %, green line) with respect to the elapsed time.**

**Table 8. Mean $D_{max}$ for each under- (S1), pre- (S2), and super-saturated (S3) stage for NaCl and CaCl$_2$ powder experiments.**

| Stage | NaCl Exp. #1 | NaCl Exp. #2 | CaCl$_2$ Exp. #1 | CaCl$_2$ Exp. #2 |
|-------|--------------|--------------|------------------|------------------|
| S1 | 26.45 μm | 26.55 μm | 20.17 μm | 18.62 μm |
| S2 | 43.17 μm | 49.17 μm | 38.45 μm | 40.36 μm |
| S3 | 39.73 μm | 45.09 μm | 38.35 μm | 45.62 μm |

## 5 Conclusions

455    In this study, we compared the particle (PSD, $F_{act}$, and $D_{act}$) and cloud droplet growth (DSD, $D_{mean}$, and $D_{max}$) characteristics of NaCl and CaCl$_2$ powders, which are used for warm-cloud seeding experiments conducted in South Korea, using the K-CPEC. The PSD characteristics of both the powders showed a bimodal size distribution of 11 nm–17 μm. The peaks of the first mode for each powder were 0.19 and 0.24 μm, respectively, while the peaks of the second mode were the same at 1.84 μm. In other words, since NaCl powder contains more smaller particles than CaCl$_2$ powder, cloud seeding should be performed

460    in a relatively more supersaturated environment to increase the fraction of CCN activation. In this study, the maximum cooling rate of the cloud chamber was –7.31 K min$^{-1}$ (mean: –2.27 K min$^{-1}$) and the maximum updraft velocity was 18.35 m s$^{-1}$ (mean: 13.52 m s$^{-1}$) in NaCl Exp. #2. As the air inside the inner chamber was evacuated using a vacuum pump, the residence time of the droplets in the chamber would be much shorter than that under natural conditions. Therefore, the experiments conducted in this study could not sufficiently achieve droplet-growth through collision and coalescence. Nevertheless, the droplet

465    diameters varied from 1 μm (in the S1 stage) to 90 μm (in the S3 stage), denoting the size range for drizzle droplet formation. In the case of NaCl, pre-deliquescence hygroscopic growth occurred under RH conditions below the DRH (75 %), followed by an active deliquescence transition near the DRH. Post-deliquescence hygroscopic growth was observed at RH exceeded 85 %, and cloud droplet formation occurred through condensational growth under supersaturated conditions (RH > 100 %). For CaCl$_2$, which has a lower DRH (28 %), the deliquescence transition was already activated under the initial experimental

470    condition (RH < 60 %). Therefore, CaCl$_2$ particles were larger than NaCl particles at the beginning of the experiment. Consequently, cloud droplet formation began at an earlier phase in the CaCl$_2$ experiments than in the NaCl experiments, with droplet growth initiated approximately 49 s and 52 s earlier in Exp. #1 and Exp. #2, respectively. However, compared with the CaCl$_2$ experiments, the NaCl experiments resulted in larger droplets, showing $D_{mean}$ values 2–3 μm greater and $D_{max}$ values of 65.71 μm in Exp. #1 and 89.16 μm in Exp. #2. Experiment #1 followed a wet adiabatic lapse rate (4.26–4.82 K km$^{-1}$) under supersaturated conditions, while experiment #2 adhered to a stable environmental lapse rate (2.28–2.42 K km$^{-1}$). These results

475    demonstrate that the cloud chamber experiment conducted in this study is capable of simulating a range of atmospheric

conditions, including natural environments and forced uplift scenarios, such as those induced by orographic effects, frontal systems, and other dynamic processes. Therefore, these chamber experiments can aid in clearly understanding the characteristics of cloud seeding materials. Based on this, it is expected that developing strategies for warm-cloud seeding experiments will improve the effectiveness and efficiency of cloud seeding.

**Author contributions:** BYK, MB, and JWC designed the experiments. BYK and MB conducted the experiments and analyzed the results. BYK performed the methodology development, data collection, programming, visualization, and investigation. BYK primarily wrote the manuscript, with MB and JWC contributing to review and editing. YK provided resources, and SK was responsible for project administration.

**Data availability:** All data can be provided by the corresponding authors upon request.

**Acknowledgements:** This work was funded by the Korea Meteorological Administration Research and Development Program "Research on Weather Modification and Cloud Physics" under Grant (KMA2018-00224).

**Competing interests:** The authors declare that they have no conflict of interest.

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
