# Peer review of "Analysis of hygroscopic cloud seeding materials using the Korea Cloud Physics Experimental Chamber (K-CPEC): A case study for powder-type sodium chloride and calcium chloride"

_EGUsphere, 2025_

## Author Response (AR1)

**Reviewer #1**

We would like to note that this comment appears to be identical to the one provided in the previous round of review. Since it was already addressed during that pre-review stage, it has not been included in the current revision. We thank the reviewer for their careful review and thank the editor as well.

For clarity, we reiterate our previous response here:

In the manuscript submitted by Kim et al., the authors built a Korea Cloud Physics Experiment Chamber and studied the hygroscopicity and cloud activation ability of two common salts used in warm cloud seeding experiments. I suggest this paper fits well within the scope of AMT. Nevertheless, I have a few comments for the authors before its acceptance for publication.

Thank you for reviewing our manuscript. We sincerely appreciate the reviewer's thoughtful and constructive comments. In response, we have thoroughly revised the manuscript to address both the major and specific comments raised. We believe these revisions have improved the overall quality and enhanced the clarity of the manuscript.

Major comments:
1. In the introduction section, the author highlighted the importance of cloud seeding in precipitation production. However, the outcomes of cloud seeding are not always positive and there is more literature reporting that cloud seeding fails to produce precipitation efficiently. The authors should also point this out and summarize the related studies.

We have added the sentences as follows:

L43–46: Cloud seeding can shorten the duration of the precipitation process and increase precipitation intensity. However, it may not always result in more rainfall than natural precipitation, depending on meteorological conditions (Silverman, 2003). Therefore, proper assessment of meteorological conditions and appropriate seeding strategies are essential for effective cloud seeding.

2. The authors tested two types of powder, NaCl and CaCl2. However, they didn't claim the motivation to use the two materials in warm cloud seeding. What are the pros and cons of using these materials compared to other materials?

We have added the sentences as follows:

L54–58: The flare-type operation disperses seeding materials at a constant rate once ignited, limiting the ability to adjust the quantity during cloud seeding experiments. The powder-type operation—provided that sufficient cargo space is available within the aircraft—allows for more precise control over both the quantity and rate of seeding. It also accommodates a wider range of seeding agents and is relatively more cost-effective than flare-type agents.

3. One of the main conclusions from the authors' findings is that "The CaCl2 powder, with strong deliquescence, exhibited a high cloud condensation nuclei (CCN) activation fraction in low-RH conditions; the NaCl powder, with high hygroscopicity, produced larger cloud droplets under supersaturation conditions." It is important to note that the two types of particles tested in this study differ in size, which significantly influences CCN activation and hygroscopicity. As a result, the comparison may not be an 'apple-to-apple' case. The authors may consider using alternative metrics instead of Fact to quantify the hygroscopicity of the produced aerosol, such as the kappa value, which accounts for particle size.

**We have revised and added the sentences as follows:**

**L17–21: NaCl and CaCl$_2$ powders showed distinct particle growth behaviors owing to the differences in their deliquescence and hygroscopicity. The rate of cloud droplet formation in the NaCl powder experiments was slower than that for CaCl$_2$; however, the mean and maximum droplet diameters were approximately 2–3 µm and 10–20 µm larger, respectively. The droplet diameter varied from 1 to 90 µm, and large cloud droplets (30–50 µm) that served as the basis for drizzle embryo formation were also observed.**

**L191–194: NaCl and CaCl$_2$ have different hygroscopicity parameters (κ = 1.24 and 0.78, respectively) and deliquescence relative humidities (DRH = 75 % and 28 %, respectively) (Fountoukis and Nenes, 2007; Liu et al., 2014). Consequently, they are expected to exhibit different deliquescence transitions and hygroscopic growth behaviors during cloud chamber experiments.**

**L385–388: Therefore, before the RH reached 85 %, the mean particle size within the size distribution below 2.84 µm was 1.06 µm, which was larger than the mean particle size of 0.94 µm in the NaCl experiments. In addition, the mean number concentration was approximately 692 cm$^{-3}$, which was 45 % higher than that in the NaCl experiments (Xueling et al., 2021; Peng et al., 2022).**

4. It is unclear how the authors define CCN in their cloud chamber experiments. For example, in Line 304, they stated that "Immediately after the beginning of the experiment, approximately 56 % of the number concentration was measured as CCNs (D > 0.3 µm) compared to the injected total number concentration under conditions of RH 55 %. ". Did the authors define particles that can grow beyond 0.3 µm as CCNs? If so, what is the rationale behind this definition?

**We have revised the sentences as follows:**

**L331–338: Immediately after the beginning of the experiment, particles larger than 0.3 µm accounted for approximately 56 % of the total number concentration for an RH of approximately 55 %. This figure was 22 % higher than the proportion of the number concentration (approximately 34 %) for particles larger than 0.3 µm measured in the aerosol chamber experiment with NaCl powder. Initially, the RH remained below the DRH, and thus, the deliquescence transition was not activated. However, owing to the strong hygroscopicity of NaCl, water vapor might have been taken up even below the DRH, leading to pre-deliquescence hygroscopic growth characterized by the formation of a thin water layer (Tang and Munkelwitz, 1993).**

**L380–385: Immediately after the beginning of the experiment, the number concentration of**

**particles larger than 0.3 µm was approximately 80 % compared with the injected total number concentration under an RH of approximately 58 %. This figure is 27 % higher than the proportion of number concentration (approximately 53 %) for particles larger than 0.3 µm measured in the aerosol chamber experiment with CaCl₂ powder. Owing to the low DRH of CaCl₂ (28 %), a large number of particles might have undergone deliquescence transition immediately upon injecting CaCl₂ powder into the cloud chamber.**

5. One of the major conclusions, "The CaCl2 powder, with strong deliquescence, exhibited a high cloud condensation nuclei (CCN) activation fraction in low-RH conditions" is not well supported. Based on aerosol chamber experiments, the Fact of CaCl2 was higher than NaCl at S of 0.1%. This means CaCl2 also has a stronger CCN activation ability at supersaturation conditions, not only at low RH conditions.

**We agree that the statement was not sufficiently supported, and we have removed it from the abstract accordingly.**

Also, the authors seem to equate the hygroscopic behavior of CaCl₂ at RH < 100% with CCN activation. However, CCN activation typically refers to particle growth under supersaturation (RH > 100%), whereas CaCl₂ undergoes deliquescence at lower RH levels. Could the authors clarify this distinction? For example, in Line 305: "This result contrasts with the higher Fact of approximately 77 % measured under S = 0.1 % condition, as fewer particles were activated as CCNs in low RH conditions. "

**The overall description of the NaCl and CaCl₂ experiments in Section 4.2 has been revised to describe particle growth behavior across different RH conditions, with key findings summarized in the conclusion (Section 5) as follows:**

**L453–463: In the case of NaCl, pre-deliquescence hygroscopic growth occurred under RH conditions below the DRH (75 %), followed by an active deliquescence transition near the DRH. Post-deliquescence hygroscopic growth was observed at RH exceeded 85 %, and cloud droplet formation occurred through condensational growth under supersaturated conditions (RH > 100 %). For CaCl₂, which has a lower DRH (28 %), the deliquescence transition was already activated under the initial experimental condition (RH < 60 %). Therefore, CaCl₂ particles were larger than NaCl particles at the beginning of the experiment. Consequently, cloud droplet formation began at an earlier phase in the CaCl₂ experiments than in the NaCl experiments, with droplet growth initiated approximately 49 s and 52 s earlier in Exp. #1 and Exp. #2, respectively. However, compared with the CaCl₂ experiments, the NaCl experiments resulted in larger droplets, showing D$_{mean}$ values 2–3 µm greater and D$_{max}$ values of 65.71 µm in Exp. #1 and 89.16 µm in Exp. #2.**

6. The authors also demonstrated that "as the air in the inner chamber was evacuated using a vacuum pump, the CCNs or droplets in the chamber may have been lost". Have the authors conducted experiments to quantify wall losses in aerosol and cloud chambers?

**We appreciate the reviewer's insightful comment. Currently, our K-CPEC facility does not support CN measurements with the CPC under low-pressure conditions, which makes it difficult to directly quantify aerosol losses during the cloud chamber experiments. We acknowledge this limitation and are working to improve our measurements accordingly.**

Specific comments:

1. What do authors mean by "until supersaturation (RH>100%) and droplet formation" (L15).

**We have revised the sentence as follows:**

**L15–17: The experiments were initiated at low RH (< 60%), and the variations in the cloud droplet concentration and diameter were observed as RH increased, leading to supersaturation (RH > 100%) and subsequent cloud droplet formation.**

2. Quantitative results should be given in the abstract. For example, "large cloud droplets that served as…"; "low RH-conditions"; "supersaturation conditions".

**We have added the quantitative figures as follows:**

**L18–21: The rate of cloud droplet formation in the NaCl powder experiments was slower than that for CaCl₂; however, the mean and maximum droplet diameters were approximately 2–3 μm and 10–20 μm larger, respectively. The droplet diameter varied from 1 to 90 μm, and large cloud droplets (30–50 μm) that served as the basis for drizzle embryo formation were also observed.**

3. Section 2.2. Can authors comment on the homogeneity of temperature within their aerosol and cloud chambers? As the temperature of the chamber is only reported from few thermal couples located at four different places, would these temperature uncertainties cause the bias of the supersaturation calculation? How much would this be?

**We have added the sentence and reference as follows:**

**L242–244: During both experiments, the temperature, pressure, and relative humidity in the aerosol chamber were controlled, with mean standard deviations of 0.05 °C, 4.73 hPa, and 0.08 %, respectively.**

**L149–152: The relative humidity (RH) inside the inner chamber was calculated using the dew point temperature measured by a chilled mirror hygrometer (Buck Research 1011C), which drew air from the inner chamber for measurement, and the air temperature measured by the thermocouple (Buck Research Instruments, 2009).**

**The standard deviation of the wall and air temperatures measured from four directions in the cloud chamber experiment was ±0.2°C (L211–212). In this study, relative humidity (RH) was calculated using Buck's formula (Buck Research Instruments 1011C manual, 1996), based on the dewpoint temperature measured by a hygrometer (accuracy ±0.1°C) and the air temperature measured by a thermocouple (accuracy ±0.2°C).**

4. L169: Can authors briefly introduce this air jet milling process?

**We have added the sentence as follows:**

**L185–186: Air jet milling produces fine powders by inducing particle–particle collisions through**

**high-velocity compressed air (Kou et al., 2017).**

5. L191: Can authors schematically show the location of the four thermocouples in the inner chamber?

**We have added red markers to indicate the locations of the thermocouples in Fig. 1.**

6. L211: Does the "remaining number concentration of aerosol" refer to aerosol remaining from the last cleaning procedure or last chamber experiments? The "experiment was performed once more" means the cleaning procedure will be performed once more, or the chamber experiment?

**We have revised the sentence as follows:**

**L231–232: If the remaining number concentration of the aerosol was more than 10 cm$^{-3}$, the cleaning procedure was performed once more.**

7. L225: step 5. Was the aerosol in the cloud chamber injected from the aerosol chamber or from the aerosol generator?

**We have revised the sentences as follows:**

**L225–226: Step 4: Inject the aerosol into the aerosol chamber using an aerosol generator**

**L247–248: Step 5: Inject the aerosol into the cloud chamber using an aerosol generator**

8. Section 4.1: SMPS measures the mobility diameter, while OPC gives the optical diameter of aerosol particles. Do authors assume they are the same, or how do they merge them? Please clarify this.

**We have added the sentence as follows:**

**L237–239: The DMA of the SMPS determines particle size based on electrical mobility, whereas the OPC measures the optical diameter, which depends on the refractive index (RI) of the particle. Accordingly, the size parameter of OPC was set based on the RI of 1.54 for NaCl and CaCl$_2$ (Zinke et al., 2022).**

9. L255: The particle size in real-world seeding events can be smaller also because the greater coagulation process within the chamber experiments due to lower wind speed. You can also cite:

Critical Size of Silver Iodide Containing Glaciogenic Cloud Seeding Particles
Jie Chen, Carolin Rösch, Michael Rösch, Aleksei Shilin, and Zamin A. Kanji

**We have added the sentence as follows:**

**L278–280: In aerosol chamber experiments, high aerosol concentrations combined with prolonged residence times may promote particle coagulation (Chen et al., 2023). However, in this study, the total number concentration during both experiments ranged from 1100 to 1200 cm$^{-3}$, suggesting that the effect of coagulation on the PSD was minimal.**

10. L270: What's the critical size for NaCl to activate at S of 0.1%? Can you find this size in the literature to validate your conclusion?

**We have added the sentence as follows:**

**L298–300: In this study, since $D_{act}$ was determined using the method proposed by Hung et al. (2014), it may differ from the critical diameter ($D_{crit}$) measured using a DMA (e.g., $D_{crit}$ of NaCl = 100 ± 4 nm at S = 0.1 %; Niedermeier et al., 2008).**

11. Was the aerosol chamber experiment for each particle type conducted only once? If so, do the authors have repeated measurements to confirm the reproducibility of their results?

**We conducted multiple preliminary tests to optimize and standardize the experimental conditions for each particle type (i.e., NaCl and $CaCl_2$ powders), ensuring similar environmental settings across all chamber experiments. Through this process, we carefully controlled the experimental setup and measurement procedures to enhance the reliability and consistency of the results. Although only two representative experiments per particle type are presented in this study, we plan to perform repeated experiments in future studies to demonstrate the reproducibility of the K-CPEC system.**

**Reviewer #2**

This manuscript describes the Korea Cloud Physics Experiment Chamber (K-CPEC) as a new cloud chamber facility for developing cloud seeding technology and investigating on aerosol-cloud Interactions. The chamber is equipped with a promising set of instruments to characterize the CCN activation ability of aerosol particles and the microphysical processes with regard to cloud droplet activation. The authors present how the aerosol and the cloud chambers can be used to study warm clouds by testing powder-type hygroscopic materials applied in cloud seeding experiments. Since the deviations from a ideal expansion process for both air pressure and temperature may affect CCN activation and cloud droplet growth inside the inner chamber, further consideration will be needed to evaluate the performance of these concept experiments and discuss on validating the results.

In the aerosol chamber experiments on the characteristics of NaCl and CaCl2 powders, the effects of PSD should be considered. It is speculated that the difference in the activation fraction of CCNs at low supersaturation (0.1 %) seems to be due to the difference in mode diameter. The gaps in the activation diameter between two sample powders was quite small over the supersaturation range (0.1–1 %), so it is unclear whether it is possible to distinguish the suitable environments for each powder in cloud seeding field experiments as proposed in this paper.

Regarding the cloud chamber experiments on the observations of cloud droplet formation, it is essential to discuss phenomena occurring under water sub-saturation and supersaturation separately so that the onset of cloud droplet formation and the measured RH at those time should be clarified. The existence of cloud droplets at the under-saturated stage (RH $\leq$ 85 %) and pre-saturated stage (85 % < RH $\leq$ 100 %) are questionable without noteworthy explanation. When discussing measurements under sub-saturated conditions, it is necessary to explain how to identify pre- and post-deliquescent salt particles. The main discussion should be focused on how such salt particles lead to their CCN activation and readily grow into larger cloud droplet up to drizzle embryo size at the super-saturated stage. From this perspective, it is necessary to consistently clarify the relationship between the deliquescence and hygroscopic properties of each salt particle type and the number concentration and size distribution of the induced cloud droplets.

**Thank you for carefully reviewing our manuscript. We truly appreciate the reviewer's detailed and constructive comments. In response, we have thoroughly revised the manuscript to address both the major and specific comments raised. We believe these revisions have improved the overall quality and enhanced the clarity of the manuscript.**

**Major comments:**

L20-21: Regarding the notation "The droplet diameter varied from 1 to 90 μm", it may include a reflection of size changes due to deliquescence, especially for particles of a few microns in size. It is not clear how to distinguish between aerosols and aqueous solutions, so for "The droplet diameter", how about expressing it as "the particle diameter including aerosols and droplets"?

**We have revised this sentence as follows:**

**L20–21: The particle diameter, including aerosols and droplets, varied from 1 to 90 μm, and large cloud droplets (30–50 μm) that served as the basis for drizzle embryo formation were also observed.**

L258-260: Regarding "In the cloud chamber experiment, the particles measured by the OPC were assumed to be water droplets. Therefore, the size parameter of the OPC was set based on the RI value of 1.33 (water).", the timing of onset of CCN activation is a critical issue in the cloud chamber experiment, and so it is incomprehensible to treat all OPC measurement data in the sub-saturated region as water droplets and to compare and discuss the cloud droplet formation process of different materials without applying the specific method of distinguishing aerosols from cloud droplets. How should we interpret the identification of water droplet formation in the sub-saturated region?

**When the relative humidity is below supersaturation, CaCl₂, which has a low deliquescence relative humidity (DRH, i.e., 28%), can already dissolve and exist as an aqueous solution suspended within the chamber under the initial experimental conditions. In contrast, NaCl only deliquesces above a higher RH threshold (i.e., 75%). Notably, both aerosols and droplets can coexist even below or above the DRH, depending on the conditions. The deliquescence process of a substance can vary depending on factors such as temperature, turbulence, and the shape and size of the particles.**

**In the present study, it was challenging to clearly distinguish the phase transition points of the two substances based solely on observations. Therefore, DRH values reported in previous studies were used to determine whether deliquescence had occurred. In addition, droplet-like (spherical) particles were identified through CPI observations. However, due to the observational limitations of the OPC, it was challenging to differentiate between phase states. To ensure continuity in the analysis, the refractive index was set to 1.33.**

**We anticipate that future studies utilizing polarization data from the Cloud Aerosol Spectrometer with Depolarization (CAS-DPOL, DMT) will enable the distinction between droplets and solid particles. At present, we are improving the design of the mounting module to be installed at the bottom of the internal chamber to facilitate precise measurements with CAS-DPOL.**

L306-309: For S ≥ 0.2%, some $D_{act}$ values for NaCl powder are relatively smaller than those for CaCl2 powder, so it is not possible to say clearly about the differences in target clouds in field cloud seeding experiments. It should be noted that the results depend on the PSD and mode diameter of the sample particles used in this study. Because the PSD in the aerosol chamber changes with time, and micron-sized particles in particular fall out quickly, using the initial PSD value will affect the calculation of $D_{act}$ values for higher S. Was the updated PSD data applied to the calculation of the $D_{act}$ values?

**We agree with the reviewer's comment. In the absence of aerodynamic flow, relatively large particles (> 10 μm) can undergo dry deposition. To address this, a fan was operated inside the aerosol chamber to ensure homogeneous distribution of aerosols and to keep relatively large particles suspended. Aerosol observations commenced immediately after injection of the target aerosol. Therefore, the $D_{act}$ (also, $F_{act}$) was calculated based on the mean particle size distribution (PSD) observed within one hour from the start of the measurement.**

**We have revised the sentence as follows:**

**L307–308: When S was 0.1 %, the $D_{act}$ of the NaCl powder was 135.8 nm and that of CaCl₂ powder was 126.3 nm, with a particle size difference of approximately 10 nm.**

L347-351: In OPC measurements, the RI value of 1.33 (water) is applied even in the unsaturated region. Unless it is any evidence that the particles observed during the deliquescence transition are cloud droplets, the particle size measurements may not be accurate. If the cloud droplets (several tens of μm in size) measured with the CPI under sub-saturated environment are correct, the spatial representativeness of the RH measurement or the accuracy of the RH values is required to be precise (is RH>100% unevenly distributed?). Regarding Figure 4i, in the other three cases, the timing at which the CPI first measured was at 85% RH (near the vertical blue lines), but is it relatively earlier?

**We have added image samples captured by the CPI in Figs. 4i, j, and 5i, j. In NaCl Exp. #1, particles captured at the beginning are assumed to be aerosols with non-spherical shapes. After the onset of deliquescence, they were observed as water droplets exhibiting droplet-like (spherical) shapes. In NaCl Exp. #2, the experiment progressed more rapidly than in Exp. #1; however, similar to Exp. #1, aerosols and droplets were distinguishable in the CPI images taken before and after deliquescence. In $CaCl_2$ Exp. #1 and #2, water droplets with droplet-like shapes were observed from the beginning of the experiment. This suggests that relatively large aerosols may form droplets even at comparatively low RH (< 85 %). Meanwhile, relatively small aerosols began to exhibit hygroscopic growth around 85 % RH.**

**We have revised the sentence as follows:**

**L430–432 and L442–444: (i, j) presents the plots for size-resolved number concentration, mean diameter ($D_{mean}$, black line), and maximum diameter ($D_{max}$, red line), with the horizontal gray dashed lines indicating diameters of 30 and 50 μm, respectively, and the images present particles captured by the CPI under RH < 85 % conditions.**

L351-354 Isn't "the most significant decrease in mean absolute humidity" a contradiction to the increase in RH? Also, when RH exceeds 100%, is the transition to condensational growth unclear?

**The absolute humidity measured by the LI-COR decreased, whereas the RH tended to increase due to changes in air temperature ($T_{air}$) and dew point temperature ($T_{dew}$). The RH increased more slowly in this section compared to in other observation periods. When the RH exceeds 100 %, it indicates that condensation growth is clearly occurring.**

L469-482: Since the definition of cloud droplets and the timing of their generation (onset) in this study are not clear, it is difficult to understand the results that cloud droplets were induced even in the S2 stage (unsaturated region). If the particles captured by CPI in the S2 stage are cloud droplets, how can their formation process be explained? Considering the relationship between the temperature lapse rate and the updraft velocity in the atmosphere, is what the authors described here about the applicable atmospheric conditions as the scope of these experimental setup appropriate?

**We have added CPI images to Figs. 4i, j and 5i, j. All images captured after deliquescence show spherical-shaped water droplets. Exp. #1 appears to follow the moist adiabatic lapse rate, resembling conditions in the actual atmosphere, whereas Exp. #2 appears to follow a relatively stable environmental lapse rate. In the S2 stage, unlike in other sections, absolute humidity exhibited a significant decreasing trend, and particle size change was also clearly observed in the**

**OPC.**

Specific comments:

L67-70: Considering the timing of conducting cloud seeding experiments for suitable warm clouds, the occurrence characteristics of each cloud type in each season should also be described.

**We have added the sentence as follows:**

**L70–72: During the period from spring to fall, including the rainy season, similar cloud frequencies (Sc: 63 %, Cu: 15 %, As: 15 %, and Ac: 7 %), cloud top heights (low-altitude: 2.45 km and middle-altitude: 3.26 km), and relatively high cloud top temperatures (low-altitude: 4.01 °C and middle-altitude: 0.61 °C) were observed.**

L122-124: How long does it take for the heat transfer fluid to pass from inlet to outlet inside the cloud chamber? Also, what is the approximate elapsed time for the heat transfer fluid to circulate through the entire path of the cooling system?

**The inner chamber of the cloud chamber has an octagonal prism shape and is composed of 21 segments in total: 2 at the top, 2 at the bottom, 8 on the upper plate, and 9 on the lower plate (the door part of the inner chamber consists of two half-sized segments). The flow rate for each segment is 2.75 m$^3$ h$^{-1}$ (1.38 m$^3$ h$^{-1}$ for the door), and the time required for the heat transfer fluid to pass through a single segment completely is approximately 7 s. The total circulation time for the heat transfer fluid through the entire cooling system and chamber of the K-CPEC facility is approximately 98 s.**

**We have revised the sentences as follows:**

**L118–121: The inner chamber is made from stainless steel (4 mm thick), and the inside wall of the inner chamber has a meandering pattern with copper pipes (29 mm in diameter) passing from the left to right (at intervals of 40 mm) through each panel of the octagonal prism structure (21 segments in total: 2 for the floor, 2 for the ceiling, 9 for the lower walls—including 2 segments forming the front door—and 8 for the upper walls).**

**L123–126: The heat transfer fluid was circulated continuously at a flow rate of 55 m$^3$ h$^{-1}$ through the copper pipes of all the walls of the inner chamber (with an approximate residence time of 7 s per wall segment) and the stainless-steel pipes located between the cooling system and the cloud chamber (with an approximate circulation time of 98 s), using the brine supply pump of the cooling system.**

L127-129: To what extent does the evacuation rate corresponding to the controlled flow rate with the vacuum pump and the opening rate of the SV cover the range of the updraft velocity(m/s)?

**It can be adjusted between 0.1 and 19 m s$^{-1}$ based on the initial pressure of 1000 hPa.**

**We have revised the sentence as follows:**

**L130–132: A solenoid valve (SV) was installed at the top of the cloud chamber to control the flow rate of the vacuum pump by adjusting the opening rate. This flow rate control can generate an updraft velocity ranging from 0.1 to 19 m s⁻¹ inside the cloud chamber, based on an initial pressure of 1000 hPa.**

L129-134: What models of a triple filter in the dry air system and the pure water system products are used? What are the humidity control ranges (min/max dew points) and how long does it take for conditions inside the chamber to reach these values?

**AIRFILTER ENGINEERING products were used. The humidity control range is 0 to 100%, and the dew point temperature range is –75 to 50 °C (see Table 2). The time required to reach a target RH varies depending on the air temperature inside the chamber. For example, under the initial conditions of this study ($T_{air} \approx 20$ °C), it takes approximately 30 min to increase RH from 0 to 60%. The lower the temperature, the faster the reference RH is reached.**

L154-155: What is the typical rate (cm³/min) at which the aerosol generator injects the aerosol into the inner chamber?

**In the present study, samples were injected into the aerosol and cloud chambers, respectively, at a rate of 80 mm h⁻¹ (400–500 cm⁻³ min⁻¹) to minimize particle loss, such as dry deposition of large particles. The number concentration may vary depending on particle size.**

L156-158: The measurement range of OPC covers not only cloud droplets but also aerosol sizes. If so, how can these particles be identified? How to calculate total cloud droplets number concentration? Similarly, for L217-218.

**As shown in Fig. 6, phase changes can be inferred from changes in size distribution; however, it is difficult to distinguish between aerosols and droplets based solely on OPC size information.**

**We have revised the sentence as follows:**

**L159–162: An optical particle counter (OPC) and a cloud particle imager (CPI) were used to measure both aerosol and droplet particles. The OPC measures particles in the 0.3–17 μm size range, whereas the CPI detects particles ranging from 10 μm to 2 mm using a high-speed camera (Connolly et al., 2007).**

**L222–224: The cloud droplets formed during this process were observed using the OPC and CPI. The cloud DSD was constructed at 1 s intervals, and the mean diameter was calculated by averaging the DSDs over the observation period using both datasets.**

L163-165: Is the CPC measurement not affected by the pressure difference between inlet and outlet? Why not sample the air like other measurement instruments?

**We are continuously improving the K-CPEC facility for chamber experiments. However, due to the absence of a low-pressure pump in the present study, CPC measurements could not be**

**performed during the experiments.**

L170-172: Describe more precisely the roles of the OPC and the SMPS in measuring aerosol size.

**This content is written in Section 4.1 L280–282.**

L208-209: I don't understand why it is necessary to mention "the growth of cloud droplets was observed at 900 and 840 s, respectively" in this paragraph.

**This indicates that the experimental duration varies depending on the SV settings.**

**We have revised this sentence as follows:**

**L213–214: In the cloud chamber experiment, for the SV values (of the vacuum pump) of 20 % and 50 %, the experiments lasted 900 and 840 s, respectively.**

L212-214: Does the $T_{wall}$ measurement positions take into account not only the geometric representative points of the inner chamber, but also the transition time of the heat transfer fluid through the wall panel of the inner chamber?

**The transfer time of the fluid was not considered because the heat transfer fluid was circulated and cooled continuously. The small standard deviation of $T_{wall}$ was attributed to the thermocouples being installed at identical positions (i.e., the midpoint of the copper tube length) in each segment, resulting in no significant difference in response time among the locations.**

L234-236: Is ΔP kept at 30 hPa during aerosol injection, or does it decrease gradually according to the injection rate without exhausting?

**A minimum pressure difference of 30 hPa was set because aerosol injection into the chamber is hindered without a pressure difference between the inside and outside air. After injection, the pressure increased slightly (4–5 hPa).**

L238-239: Are there any significant changes in CN number concentration and PSD during the 1-hour observation period?

**There was no significant difference in the distribution. Due to differences in measurement characteristics between the two observation instruments, a 1 h mean was used to obtain a representative value. Additionally, the CCN observation duration (approximately 55 min) was taken into account.**

L245-246: How was the pressure in the aerosol chamber controlled during SMPS and OPC measurements, which required a certain total sample flow rate?

**After aerosol injection, the chamber was left sealed without additional pressure control. The**

**pressure decreased gradually by approximately 20 hPa from the initial value during the SMPS, OPC, CPC, and CCN counter measurements.**

L253-254: After the humidification, was ΔP adjusted to be 30hPa, as same as the procedure in the aerosol chamber experiment?

**For cloud chamber experiments, both water vapor and aerosols must be injected; therefore, the chamber pressure is set to be 100 hPa lower than the ambient pressure.**

**We have revised the sentence as follows:**

**L259–261: The reference pressure of the cloud chamber was set to 100 hPa lower than the ambient air pressure, considering the air pressure increase caused by the supply of water vapor and aerosols.**

L258: Was there little effect on drying due to aerosol injection?

**Dry air (containing aerosols) was injected through the RBG1000 aerosol generator, but it did not affect the RH significantly because the injection duration was short and the chamber volume was sufficiently large.**

L262-263: The initial values of Tair in NaCl Exp. #1 and #2 are approximately 0.4°C lower than those of $T_{wall}$. The differences between them were relatively large compared to the two cases in CaCl2 experiment. Could this be due to some difference in the procedure?

**The experimental procedure was identical in all cases. To set the initial air temperature ($T_{air}$) before the experiment, the cooling system does not maintain a constant value but instead allows small fluctuations around the setpoint. This behavior is common in all cooling systems and falls within the measurement uncertainty of the thermocouple (±0.5 °C).**

L273: At what timing were these PSDs measured?

**We have revised the sentence as follows:**

**L280–284: These distributions were merged with those of OPC values measured after the maximum size measurable by SMPS, that is, the PSDs of 11 nm–0.48 μm were measured using SMPS and the PSDs of 0.48–17 μm were measured using OPC, with each distribution representing 1 h mean values for each size bin. Notably, the data in Fig. 2 were obtained simultaneously during the same experiment as those shown in Fig. 3 for the CPC and CCN counter measurements.**

L323-324: Is it inevitable that the operation of the SV and vacuum pump will have an effect after the experiment has started? If necessary, describe it as a part of the experimental procedure.

**We have revised the sentence as follows:**

**L256–257: Step 6: Begin the experiment and observation (with vacuum pump operation and wall temperature adjustment).**

L326-328: Excluding the transitional period up to 150 seconds after the start of each experiment, the cooling rate was almost uniform at about -2K/min. Even if there is a significant difference in updraft velocity between SV 20% and 50%, the greater the updraft velocity, the greater the deviation from adiabatic expansion, so it is unclear whether the experiment conforms to the conditions proposed here.

**We agree with the reviewer's comment. In experiments with large SV openings, the wall temperature should decrease more rapidly. We are currently working on improving the cloud chamber's wall cooling system to achieve faster temperature reduction.**

**We have revised the sentence as follows:**

**L335–337: These experimental conditions may simulate cloud seeding experiments in areas with weak-to-strong convection and strong orographic updrafts (Jensen et al., 1998; Field et al., 2001).**

L329-330: Does the time it takes for the fluid to circulate in the chamber affect the homogeneity of the wall temperature? Also, are there any adjustments being made to the flow of the heat transfer fluid into the chamber in terms of the wall temperature control?

**Since the residence time of the fluid passing through each wall segment is approximately 7 s, the wall temperature may be partially non-uniform. However, the phenomenon described in L339–340 is temporary and does not significantly affect the $T_{air}$ and $T_{wall}$ variations in the present study (L218–219). We used an optimal flow rate (55 $m^3$ $h^{-1}$) to cool the internal chamber air temperature, accounting for the heat loss that may occur during circulation through the entire chamber-cooling system of the K-CPEC facility.**

L330-331: In the case where Tair-Tdew is small (initial value of RH is high), is there a possibility that it will affect the variation in air temperature? How can it be reduced?

**We believe that even slight differences in initial relative humidity (e.g., close to 100 %) can affect the experimental results. The K-CPEC facility requires significant improvements. In future, we plan to introduce a pre-tank to mix the pre-cooled heat transfer fluid (e.g., –60°C) with the circulating fluid at a certain ratio to enable smoother temperature reduction.**

L333-337: Did the authors confirm that the PSDs of each sample introduced into the cloud chamber were the same as that measured in the aerosol chamber? Also, the number concentration and PSD change with time in the chamber experiment, but have these been taken into consideration? The same applies to lines 390-392.

**In the present study, the experiment was conducted immediately after aerosol injection into the cloud chamber. Unlike in the aerosol chamber experiment, the cloud chamber experiment started with a higher initial RH, which may have resulted in differences in the PSD compared to that sampled in the aerosol chamber. Therefore, a larger proportion of larger particles was observed**

**in the cloud chamber experiment than in the aerosol chamber experiment.**

L365-366: Since the evacuation by a vacuum pump simulates the expansion process, does it take into account the reduction (or loss) of aerosols containing CCNs due to the dilution? Is an additional consideration the loss due to falling out micron-sized CCNs and cloud particles?

**In the present study, aerosol loss rates were not considered. It is presented for reference regarding changes in number concentration.**

L369-372: Shouldn't the size distributions at least in stages S1 and S2 be aerosol and cloud droplet size distribution rather than cloud DSD since it contains aerosol particles that are not activated as CCNs? The same shall apply hereinafter.

**We have revised the sentence as follows:**

**L370–373: The growth process of cloud droplets was divided into the under-saturated stage (RH ≤ 85 %, hereinafter referred to as S1), pre-saturated stage (85 % < RH ≤ 100 %, hereinafter referred to as S2), and super-saturated stage (RH > 100 %, hereinafter referred to as S3), as shown in Fig. 6, and the cloud DSD—possibly including both aerosols and droplets—was expressed.**

L376-377: In Fig. 6a, an increase in the number concentration of particle in sizes measured with CPI can be confirmed in S3 compared to S2 and, but the difference in the "right tail of the bimodal distribution" of S2 and S3 does not seem to be clear.

**We have revised this sentence as follows:**

**L377–379: This bimodal distribution persisted until RH exceeded 100 %; in this supersaturated stage, the right tail of the bimodal distribution increased relatively, as observed in the S3 shown in Fig. 6a.**

L384-385: As for NaCl, is it possible that there are a relatively large number of particles that are slightly below the OPC detection limit (D > 0.3 μm)? If so, should it be considered that those relatively small particles act as CCNs?

**We cannot explain this clearly because SMPS observations were not available for the cloud chamber experiment. Based on OPC observations, particles with D > 0.3 μm were still observed, suggesting that these particles likely grew larger.**

**Technical corrections:**

L145: Table 2. -> Table 1.

**We have revised this number.**

L214: Tdew -> dew point temperature (Tdew)

**We have revised this word.**

L269: Table 6. Experimental conditions -> Experimental initial conditions

**We have added this word.**

L313: DDact -> Dact

**We have revised this word.**

L319: "The air pressure, air, dew point," -> "The air pressure, air temperature, dew point,"

**We have revised this sentence.**

L324-326: To clarify that this is the case for NaCl Exp. #2.

**We have revised these sentences as follows:**

**L333–335: In the NaCl Exp. #2, the maximum updraft velocity (18.3 m s$^{-1}$) could be achieved in 55 s after the beginning of the experiment; the mean updraft velocity was 13.5 m s$^{-1}$. The maximum cooling rate 85 s after the beginning of the experiment was –7.3 K min$^{-1}$; the mean cooling rate was –2.3 K min$^{-1}$.**

L359-360: Add "not only in case of NaCl Exp. #1 but also in the other three cases".

**We have revised this sentence as follows:**

**L361–362: Under super-saturated (RH > 100 %) conditions, cloud droplets of 30–50 μm in size were consistently observed not only in NaCl Exp. #1 but also in the other three experiments.**

L373: shown in Fig. 4g -> shown in Fig. 4i

**We have revised this character.**

L384-385: cloud droplets were observed -> cloud particles at below freezing point were observed

**We have revised this sentence.**

L386: (Figs. 4g and 4h) -> (Figs. 4i and 4j)

**We have revised those characters.**

L425-449: The legends in Figures 4 and 5 overlap with the plots and there are some unclear parts, so they should be corrected.

**We have adjusted the positions of the legends to improve their visibility.**

L464-466: Add "in NaCl Exp. #2"

**We have added this phrase.**

**Reviewer #3**

Summary:

This study analyzes the droplet growth and cloud formation properties of two hygroscopic compounds, NaCl and CaCl$_2$. Specifically, both compounds were analyzed for its application to cloud seeding in warm clouds. The authors conducted both aerosol experiments (CPC, CCNC) and cloud chamber measurements using the Korea Cloud Physics Experiment Chamber (K-CPEC). The authors observed smaller NaCl particles compared to CaCl$_2$ and greater growth for NaCl due to greater hygroscopic behavior. The authors concluded that cloud seeding for the analyzed compounds should be done in under more supersaturated environments to increase the fraction of CCN activation. This work provides greater insight into compounds' ability to help cloud formation using experimental techniques simulating atmospheric conditions. The results of this paper have greater implications for further cloud seeding performance and highlights the use of the K-CPEC for such studies. As a result, this work is well suited for AMT and should be published. A few clarifying questions are brought up before final publication:

**Thank you for reviewing our manuscript. We sincerely appreciate the reviewer's thoughtful and constructive comments. In response, we have thoroughly revised the manuscript to address both the major and specific comments raised. We believe these revisions have improved the overall quality and enhanced the clarity of the manuscript.**

Specific Questions:

1. Line 166: I understand that temperature was hard to control, but is there a temperature range that can be provided?

**The temperature in the aerosol chamber may vary depending on the ambient air temperature, but it was typically around 20 ± 5 °C.**

**We have revised the sentence as follows:**

**L171–173: However, because dry air was supplied during the chamber-cleaning process, the aerosol chamber was slightly cooler than the ambient air and extremely dry (RH < 1 %), with the temperature typically at 20 °C ±5 °C.**

2. Line 170-174: Was a calibration performed for the CCNC ? Past studies have calibrated the CCNC using ammonium sulfate (Rose et al., 2008) - how were the instrument SS verified as being close to the SS input into the program? If a calibration was performed, please clarify in the text and put calibration results in an SI.

**In the present study, no experimental calibration of the CCN counter was performed. Instead, the SS values were determined based on the factory-level calibration procedure provided by the manufacturer. We recognize that relying solely on these preset values may introduce some uncertainty in the actual SS, as noted by the reviewer. Therefore, we have included a discussion of the potential uncertainty (up to ±10 %) associated with the SS values in the revised manuscript.**

**L180–181: Since no experimental calibration of the CCN counter was conducted in this study, the supersaturation values at each interval may carry an uncertainty of up to ±10 % (Rose et al.,**

**2008).**

3. Line 195-198: It seems as though the $N_{CCN}/N_{CN}$ results were obtained from scanning mobility CCN analysis (SMCA) method (scanning through range of diameters using SMPS then getting ratio of the distribution) as opposed to a stepping mode method (keeping diameter constant and varying SS% in CCN) - is this correct?    If so, please clarify in the text, the authors can also cite Moore et al., 2010.

**In the present study, $D_{act}$ and $F_{act}$ were calculated according to the method described in Hung et al. (2014). While the calculation approach is conceptually similar to that of Moore et al. (2010), there is a key difference in the aerosol measurement method. Specifically, in our setup, aerosol size distributions and CCN concentrations were measured separately using SMPS and a CCN counter, respectively, rather than using a fully integrated SMCA system as in Moore et al. (2010).**

**Hung, H. M., Lu, W. J., Chen, W. N., Chang, C. C., Chou, C. C. K., & Lin, P. H. (2014). Enhancement of the hygroscopicity parameter kappa of rural aerosols in northern Taiwan by anthropogenic emissions. *Atmospheric Environment*, *84*, 78–87. https://doi.org/10.1016/j.atmosenv.2013.11.032**

General questions/comments:

1. Line 124-125: Does having openings cause for any additional contamination/compounds other than the tested ones to enter the chamber and influence results?

**While a minimal background aerosol level may exist ($< 10$ cm$^{-3}$), the cleaning procedure flushes both the inner and outer chambers of the cloud chamber simultaneously with filtered dry air, effectively minimizing contamination and ensuring that only the injected aerosols dominate during the experiment.**

2. Line 358: Authors state the degree of supersaturation can not be calculated effectively due to condensation - does this have major implications on the results of the cloud chamber (e.g., SS being 0.1% instead of 0.2%)? Are there ways to improve upon this with future work? If so, can the authors address this as a future work/implication in the conclusions

**This sentence was considered for removal during the previous pre-review process but was not reflected in the revision. We have now removed this sentence. However, as the reviewer rightly pointed out, accurate estimation of supersaturation is crucial for understanding aerosol condensation growth. In future work, we plan to estimate supersaturation using a precise water vapor concentration measurement instrument based on the tunable diode laser absorption spectroscopy (TDLAS) method, as described in Lamb et al. (2023), which provides high accuracy ($\pm 5$ %) over long path lengths and within the temperature range used in our chamber experiments. However, since this improvement is still under consideration and has not yet been implemented, it was not described in the manuscript.**

**Lamb, K. D., Harrington, J. Y., Clouser, B. W., Moyer, E. J., Sarkozy, L., Ebert, V., ... & Saathoff, H. (2023). Re-evaluating cloud chamber constraints on depositional ice growth in cirrus clouds– Part 1: Model description and sensitivity tests. *Atmospheric Chemistry and Physics*, *23*(11), 6043–**

**6064.**

References:

Rose, D., Gunthe, S. S., Mikhailov, E., Frank, G. P., Dusek, U., Andreae, M. O., & Pöschl, U. (2008). Calibration and measurement uncertainties of a continuous-flow cloud condensation nuclei counter (DMT-CCNC): CCN activation of ammonium sulfate and sodium chloride aerosol particles in theory and experiment. Atmos. Chem. Phys., 8(5), 1153-1179. https://doi.org/10.5194/acp-8-1153-2008

Moore, R. H., Nenes, A., & Medina, J. (2010). Scanning Mobility CCN Analysis—A Method for Fast Measurements of Size-Resolved CCN Distributions and Activation Kinetics. Aerosol Science and Technology, 44(10), 861-871. https://doi.org/10.1080/02786826.2010.498715

**Community comments #1**

Thank you for the suggestion. We have cited this manuscript in L396 to provide clearer support for the statement.

L394–396: Owing to the low DRH of $CaCl_2$ (28 %), a large number of particles might have undergone deliquescence transition immediately upon injecting $CaCl_2$ powder into the cloud chamber (Guo et al., 2019).

Guo, L. Y., Gu, W. J., Peng, C., Wang, W. G., Li, Y. J., Zong, T. M., Tang, Y. J., Wu, Z. J., Lin, Q. H., Ge, M. F., Zhang, G. H., Hu, M., Bi, X. H., Wang, X. M., and Tang, M. J.: A comprehensive study of hygroscopic properties of calcium- and magnesium-containing salts: implication for hygroscopicity of mineral dust and sea salt aerosols, Atmos. Chem. Phys., 19, 2115-2133, 2019.